# Multiple modes of cholesterol translocation in the human Smoothened receptor

**Prateek D Bansal[1], Maia Kinnebrew[2], Rajat Rohatgi[2,3], Diwakar Shukla[1,4,5,6]***

[1]Department of Chemical and Biomolecular Engineering, University of Illinois Urbana-Champaign, Urbana, United States; [2]Department of Biochemistry, Stanford University School of Medicine, Stanford, United States; [3]Department of Medicine, Stanford University School of Medicine, Stanford, United States; [4]Department of Chemistry, University of Illinois at Urbana-Champaign, Urbana, United States; [5]Department of Bioengineering, University of Illinois Urbana-Champaign, Urbana, United States; [6]Cancer Center at Illinois, University of Illinois at Urbana-Champaign, Urbana, United States

## eLife Assessment

In this **important** study, the authors conducted extensive sets of computational and investigations of the mechanism of cholesterol transport in the smoothened (SMO) protein. The computational component integrated multiple state-of-the-art approaches such as adaptive sampling, free energy simulations, and Markov state modeling, providing **compelling** support for the proposed mechanistic model, which is further validated with **solid** experimental mutagenesis data.

**\*For correspondence:**
diwakar@illinois.edu

**Abstract** Smoothened (SMO), a member of the G-protein-coupled receptor superfamily, mediates Hedgehog signaling and is linked to cancer and birth defects. SMO responds to accessible cholesterol in the ciliary membrane, translocating it via a longitudinal tunnel to its extracellular domain. Reaching a complete mechanistic understanding of the cholesterol translocation process would help in the development of cancer therapies. Experimental data suggest two modes of translocation to support entry of cholesterol from outer and inner membrane leaflets, but the exact mechanism of translocation remains unclear. Using atomistic molecular dynamics simulations (~2 ms simulations) and biochemical assays of SMO mutants, we assess the energetic feasibilities of the two modes. We show that the highest energetic barrier for cholesterol translocation from the outer leaflet is lower than that from the inner leaflet. Mutagenesis experiments and complementary simulations of SMO mutants validate the role of critical amino acid residues along the translocation pathways. Our data suggests that cholesterol can take either pathway to enter SMO, thus explaining experimental observations in the literature. Thus, our results illuminate the energetics and provide a first molecular description of cholesterol translocation in SMO.

## Introduction

Smoothened (SMO) is a member of the G-protein-coupled receptor (GPCR) superfamily with a typical heptahelical transmembrane fold (*Wang et al., 2013*; *Wang et al., 2014*). SMO has been identified as an oncoprotein, and mutations that cause overactivity of SMO drive tumorigenesis in basal cell carcinoma and medulloblastoma (*Raleigh and Reiter, 2019*). SMO antagonists are validated

anti-cancer drugs (*Axelson et al., 2013*; *Jain et al., 2017*) but are limited by drug resistance and side effects (*Meani et al., 2014*). SMO functions as a transmembrane signal transducer in the Hedgehog (HH) signaling pathway, which is known to mediate cell differentiation during embryonic development (*Nüsslein-Volhard and Wieschaus, 1980*; *Ingham, 2022*). It transduces signals across the cell membrane, particularly in the primary cilia (*Rohatgi et al., 2007*; *Corbit et al., 2005*). However, the mechanism of endogenous activation of SMO is still a debated question in the field. SMO activation has been linked to membrane sterols in numerous studies (*Cooper et al., 2003*; *Stottmann et al., 2011*; *Nachtergaele et al., 2012*; *Myers et al., 2013*; *Blassberg et al., 2016*; *Luchetti et al., 2016*; *Huang et al., 2016*; *Byrne et al., 2016*; *Myers et al., 2017*; *Kinnebrew et al., 2019*). Cholesterol, a steroidal molecule found abundantly in the plasma membranes of vertebrates, has been identified as the endogenous agonist for SMO (*Luchetti et al., 2016*; *Huang et al., 2016*; *Byrne et al., 2016*). Membrane cholesterol has been shown to interact with and modulate GPCR activity (*Kumar and Chattopadhyay, 2020*; *Kumar et al., 2021*; *Serdiuk et al., 2022*), but in the case of SMO, cholesterol has been uniquely shown to be necessary and sufficient for SMO activation (*Luchetti et al., 2016*; *Huang et al., 2016*). Recent work has shown that SMO is activated by a minor pool of membrane cholesterol, termed accessible cholesterol, at the primary cilium (*Kinnebrew et al., 2019*; *Radhakrishnan et al., 2020*; *Kinnebrew et al., 2021*; *Siebold and Rohatgi, 2023*). Patched-1 (PTCH1), a 12-pass transmembrane protein, sits directly upstream of SMO in the HH pathway and inhibits SMO (*Ingham et al., 1991*; *Denef et al., 2000*; *Ingham et al., 2000*; *Taipale et al., 2002*). PTCH1 modulates the availability of accessible cholesterol at the primary cilium, thereby regulating SMO, with models suggesting effects on both the CRD and 7TM pockets (*Kinnebrew et al., 2019*; *Radhakrishnan et al., 2020*; *Siebold and Rohatgi, 2023*).

The mechanism by which SMO is activated has been the subject of much discussion over recent years, with multiple studies theorizing the mechanism of PTCH1's inhibition on SMO. Previous studies have suggested that PTCH1 could function as a sterol transporter (*Davies et al., 2000*; *Taipale et al., 2002*; *Bidet et al., 2011*; *Zhang et al., 2018*; *Kinnebrew et al., 2019*; *Kinnebrew et al., 2021*). A recent study using coarse-grained Molecular Dynamics simulations investigated the possibility of this process, concluding that the overall process might occur at an energetic cost of ~3–5 kcal/mol (*Ansell et al., 2023*). Recently resolved structures of PTCH1 *Gong et al., 2018*; *Qi et al., 2018*; *Rudolf et al., 2019*; *Qi et al., 2019a*; *Zhang et al., 2020* have reported the presence of a Sterol Binding Domain (SBD) and a hydrophobic conduit that extends from the outer leaflet to the extracellular space. Heterologous expression of PTCH1 in membranes reduces outer leaflet cholesterol accessibility, suggesting that PTCH1 reduces the access of SMO to membrane cholesterol (*Kinnebrew et al., 2021*).

The CRD of SMO contains a binding site for steroidal molecules (*Nachtergaele et al., 2012*; *Nachtergaele et al., 2013*; *Nedelcu et al., 2013*; *Myers et al., 2013*). The endogenous agonist cholesterol (*Byrne et al., 2016*), as well as the naturally occurring alkaloidal agonist cyclopamine (*Nachtergaele et al., 2013*; *Huang et al., 2016*), bind in the CRD. In addition, SMO has a pocket in the Transmembrane Domain (TMD), which is known to bind to multiple antagonists such as LY2940680 (*Wang et al., 2013*), SANT1 and AntaXV (*Wang et al., 2014*), cyclopamine (*Weierstall et al., 2014*), TC114 (*Zhang et al., 2017*), Vismodegib (*Byrne et al., 2016*), the synthetic agonist SAG (and variants SAG1.3, SAG1.5, SAG21k) (*Wang et al., 2014*; *Qi et al., 2020*; *Deshpande et al., 2019*; *Kinnebrew et al., 2022*), the steroidal agonists 24 S,25-epoxy cholesterol (*Qi et al., 2019b*), and cholesterol (*Deshpande et al., 2019*; *Qi et al., 2020*).

While cholesterol binding to both the TMD and CRD sites is required for full SMO activation, our work focuses on how cholesterol gains access to the CRD site, perched above the outer leaflet of the membrane (*Luchetti et al., 2016*; *Kinnebrew et al., 2022*). Multiple lines of evidence suggest that PTCH1-regulated cholesterol binding to the CRD plays an instructive role in SMO regulation both in cells and animals. Mutations in residues predicted to make hydrogen bonds with the hydroxyl group of cholesterol bound to the CRD reduced both the potency and efficacy of SHH in cellular signaling assays (*Kinnebrew et al., 2022*; *Byrne et al., 2016*) and, more importantly, eliminated HH signaling in mouse embryos (*Xiao et al., 2017*). Experiments using both covalent and photocrosslinkable sterol probes in live cells directly show that PTCH1 activity reduces sterol access to the CRD (*Kinnebrew et al., 2022*; *Xiao et al., 2017*). Notably, our simulations evaluate a path of cholesterol translocation that includes both the TMD and CRD sites: cholesterol first enters the 7-transmembrane domain bundle from the membrane; it then engages the TMD site before continuing along a conduit to

the CRD site. Thus, we analyze translocation energetics and residue-level contacts along a path that includes both the TMD and the CRD.

*Huang et al., 2018* theorized that SMO's activation involved the translocation of cholesterol from the membrane via a hydrophobic conduit to the binding site in the CRD. This study resolved the structure of active *Xenopus laevis* SMO, which showed outward movements of transmembrane helices 5 and 6 (TM5-TM6) on the intracellular end, which opened a cavity in SMO extending to the inner leaflet laterally, between TM5 and TM6. This led to the hypothesis that the entry of cholesterol into SMO could happen from the inner leaflet of the membrane, between TM5 and TM6. There is also additional evidence showing that the activity of SMO is regulated by cholesterol in the outer leaflet, which enters SMO between TM2 and TM3. *Hedger et al., 2019* reported a cholesterol binding site present at the outer leaflet, between the TM2 and TM3 helices. Using coarse-grained simulations, the authors tested a variety of membrane compositions around SMO and found that a cholesterol binding site existed in human SMO (hSMO). *Kinnebrew et al., 2021* used total internal reflection fluorescence microscopy (TIRFM) to assess the effect of PTCH1's activity on membrane cholesterol. They concluded that PTCH1 activity caused a decrease in the accessibility of cholesterol in the outer leaflet, suggesting that outer leaflet cholesterol is sensed by SMO. Furthermore, in our previous work (*Bansal et al., 2023*), we observed that cholesterol accumulated outside TM2-TM3 in the outer leaflet of inactive SMO. This was supported by the observation that when SMO was bound to its agonist SAG, we observed a tunnel opening between TM2-TM3 in the outer leaflet, which may facilitate the translocation of cholesterol into the core of SMO's TMD. We therefore hypothesize that cholesterol shows two modes of translocation to enter the TMD from the membrane - (1) starting at the outer leaflet, between TM2-TM3, or (2) starting at the inner leaflet, between TM5-TM6. Alternatively, (3) cholesterol could use both pathways if they show similar energetic behaviors.

How cholesterol moves from the membrane into the core of the TMD of SMO is still poorly understood. Cholesterol traverses SMO to ultimately reach the TMD and CRD binding sites, but the mechanism of cholesterol perception has not been elucidated yet, which gives us an opportunity to explore the mechanistic aspects of this process from both computational and experimental viewpoints. In this study, we simulate SMO by embedding it in a membrane (*Figure 1b*). We report the entire translocation path of cholesterol from the membrane to SMO's CRD for both modes of translocations (*Figure 1c*) - between TM2-TM3 in the outer leaflet of the membrane (hereafter referred to as 'Pathway 1'), and between TM5-TM6 in the inner leaflet of the membrane (hereafter referred to as 'Pathway 2'). We observe that cholesterol can translocate via both pathways, and the free energy barriers associated with Pathway 1 are lower than those of Pathway 2. We test mutations in SMO that can disrupt the movement along either pathway and show that the experimental results are further supported by simulations of cholesterol translocation in SMO mutants.

After cholesterol has entered the TMD of SMO in our simulations, we observe cholesterol moving along TM6 to the TMD-CRD interface (Common Pathway, *Figure 1d*) to access the binding site in the CRD (*Huang et al., 2018*; *Kinnebrew et al., 2022*). One of the unique features of SMO is the presence of a long helix 6 (TM6) (*Wang et al., 2013*; *Figure 1—figure supplement 1*), which acts as a connector between the CRD and the TMD. We test mutations in SMO that can disrupt cholesterol movement along this Common Pathway and show that these mutants can halt the translocation process by loss of hydrophobic contacts. These results for SMO mutants are further validated by additional MD simulations. Therefore, in this study, the entire process of cholesterol translocation was observed using aggregate 2 ms of unbiased all-atom MD simulations. Exploring the mechanism by which SMO translocates cholesterol would provide insights into the endogenous regulation of HH activity and suggest strategies for the next generation of drugs targeting SMO.

## Results and discussion
### The entry of cholesterol from the outer leaflet into the TMD exhibits the highest energetic barrier along Pathway 1

We first simulated the translocation mode in which cholesterol enters the TMD of SMO from the outer leaflet (Pathway 1). To model this process, cholesterol was placed outside of the TMD in the membrane outer leaflet, between the TM2-TM3 interface. The entry of cholesterol from the membrane into the protein was steered towards the TMD to sample the entry pathway. The frames generated were then

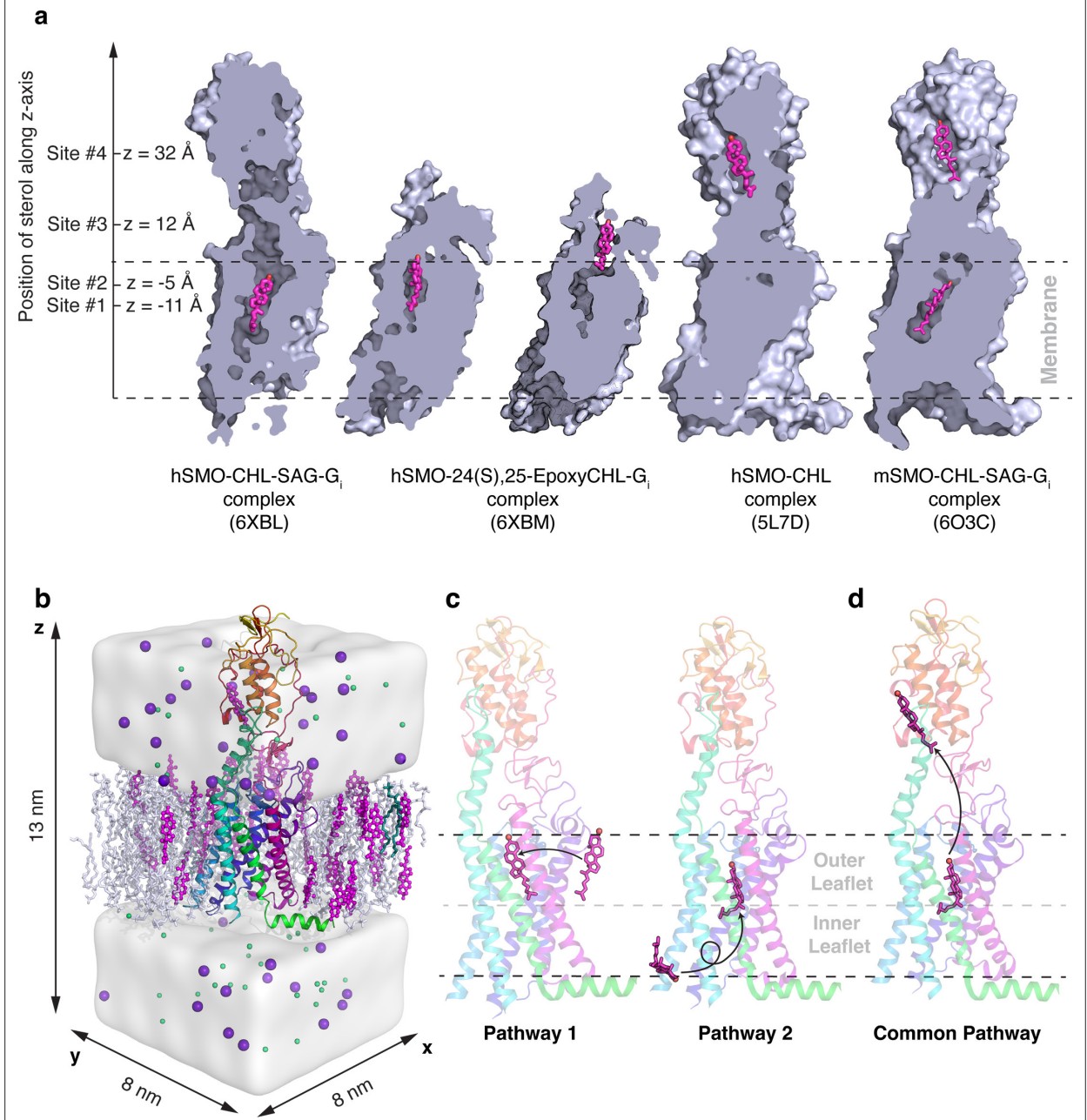

**Figure 1.** Overview of existing structures and hypotheses supporting the cholesterol translocation mechanisms in Smoothened. (**a**) The binding sites of the sterols along the hypothesized tunnel in SMO. Sterol binding sites have been identified deep in the TMD (6XBL) (*Qi et al., 2020*), at the interface of CRD and TMD (6XBM; *Qi et al., 2020*), at the CRD sterol-binding site (5L7D; *Byrne et al., 2016*), and in a dual-bound mode where cholesterol is bound to both the TMD and CRD (6O3C; *Deshpande et al., 2019*). (**b**) Example simulation system showing SMO (5L7D, cyan) embedded in a membrane (white/magenta). Water is shown as a white surface, while sodium (purple), and chloride (green) ions are shown as spheres. (**c**) Pathway 1 and Pathway 2 investigate the translocation of cholesterol from the membrane to SMO's TMD. (**d**) The Common Pathway follows the translocation of cholesterol from the TMD to the CRD. Snapshots in (**a**) are made from structures in the PDB, while (**b–d**) are frames taken from MD simulations.

The online version of this article includes the following figure supplement(s) for figure 1:

**Figure supplement 1.** Comparison of TM6 length for SMO (**A**) and *β*2AR (**B**).

used to seed unbiased simulations for adaptive sampling (see Methods section for details). This was done to estimate the energetic barriers involved in the cholesterol entry from the outer leaflet.

To enter the TMD from the membrane, cholesterol must overcome the entropic barrier of being restrained inside the protein. This can make the entry process energetically expensive. To identify the

barrier associated with entry, we projected the entire dataset on different translocation-related metrics and identified the mechanism of cholesterol translocation. The z-coordinate of cholesterol's center of mass (z-axis is perpendicular to the plane of the membrane) was used as a proxy for the progress of the overall translocation from the membrane to the CRD. The angle of cholesterol with the x-y plane (the plane of the membrane) was calculated to identify cholesterol pose. We observe that this angle (*Figure 2a* and *Figure 2—figure supplement 1a*) shows multiple minima ($\alpha/\beta$ along the Pathway 1). This provides evidence for multiple stable poses along the pathway as observed in the multiple stable poses of cholesterol in Cryo-EM structures of SMO bound to sterols (*Deshpande et al., 2019*; *Qi et al., 2019b*; *Qi et al., 2020*). A reliable estimate of the barriers comes from using the time-lagged Independent Components (tICs), which project the entire dataset along the slowest kinetic degrees of freedom. Overall, the highest barrier along Pathway 1 is ~5.8±0.7 kcal/mol, and it is associated with the entry of cholesterol into the TMD (*Figure 2—figure supplement 2*). Several factors contribute to this high barrier; first, cholesterol needs to be at the correct position - just outside the protein. In addition, cholesterol needs to be in the correct orientation - angled, with the isooctyl tail pointing towards the protein core. Additionally, the steric hindrances from the hydrophobic residues at the interface provide further obstacles for entry. Therefore, the TM2/3 residues at the entry point need to undergo conformational change to facilitate cholesterol entry – making it a rare event.

In the first stage of translocation, cholesterol is in the membrane ($\alpha$, *Figure 2*), just outside the TM2/3 helices of SMO as identified in literature (*Hedger et al., 2019*). At this cholesterol recognition site, SMO primarily contains hydrophobic residues (*Figure 2c–f*), which preferentially interact with the hydrophobic isooctyl tail of cholesterol. Therefore, the cholesterol entry involves the insertion of the tail into the TMD first, which is then followed by the hydrophilic androsterolic moiety. For the sake of clarity, we divide this entry pocket at the TM2-TM3-membrane interface into two parts - 'lower' and 'upper' pocket. The 'upper' pocket corresponds to the residues that coordinate with the androsterolic moiety of cholesterol (*Figure 2c and d*), and the 'lower' pocket corresponds to residues that lie closer to the isooctyl tail of cholesterol (*Figure 2e and f*). The upper (M286$^{ECL1}$, A289$^{ECL1}$, I$^{ECL2}$, and A$^{ECL2}$) and the lower (W281$^{2.58f}$, F391$^{ECL2}$, and M525$^{7.45f}$) pocket residues undergo conformational changes to open the space for the entry of cholesterol (*Figure 2c–f*). Here, the superscript refers to the Wang numbering scheme (*Wang et al., 2014*). This flexible motion is facilitated by G280$^{2.57f}$ in the lower pocket and G288$^{ECL1}$ in the upper pocket. Upon the entry of the cholesterol tail, an intermediate state, $\beta$ (*Figure 2d and f*) is observed where cholesterol lies flat with respect to the membrane plane and forms extensive hydrophobic contacts with A283$^{2.60f}$, I317$^{3.28f}$, F318$^{3.29f}$, and the disulfide bridge forming residues C314$^{3.25f}$ - C390$^{ECL2}$.

To reach the TMD binding site, cholesterol must first 'rock' back to its upright pose ($\alpha^*$) from its flat conformation in the state $\beta$. This rocking motion is facilitated by the polar interactions between S313$^{3.24f}$, Q284$^{2.63f}$, and the alcoholic oxygen in cholesterol. The entry of cholesterol is thus captured by a $\alpha \to \beta \to \alpha^*$ transition. For further clarity, we have plotted the minimum energy path taken by cholesterol as it translocates along this pathway (*Figure 2—figure supplement 3a, b*). To further elucidate the position of cholesterol as it enters the protein, we projected the x and y coordinates of cholesterol on a free energy landscape (*Figure 2b*, and *Figure 2—figure supplement 1a, b*). In this figure, the state $\beta$ clearly marks the transition state between cholesterol outside and inside the TMD. Overall, the angle between cholesterol and the x-y plane transitions from 90° → 0° → 90° for cholesterol to enter the protein.

To further dive into the details of cholesterol translocation, we designed mutations along Pathway 1 and measured the change in activity with respect to wild-type (WT) mouse Smoothened (mSMO). mSMO was chosen since there are no human SHH-responsive cell lines that can be passaged, edited, or transduced with genes. In addition, mSMO shows 92.8% sequence identity and 94.6% sequence similarity with hSMO for full-length sequences. Previous studies that have resolved hSMO structures have used mSMO for their structure-guided mutagenesis studies to comment on SMO activity (*Byrne et al., 2016*). Activity was measured using *Gli1* mRNA fold change in the presence of SHH (*Figure 3a*). To further validate the mutations, simulations were performed on hSMO to compute the Potential of Mean Force (PMF) of cholesterol entry in the presence of mutations (*Darve et al., 2008*; *Hénin et al., 2010*; additional details presented in Methods). Here, the PMF can characterize the barriers associated with the translocation of cholesterol, and a difference in the peak value of PMF is presented for each mutant (*Figure 3b*). We mutated G280$^{2.57f}$ to valine - G$^{2.57f}$V to test whether reducing the

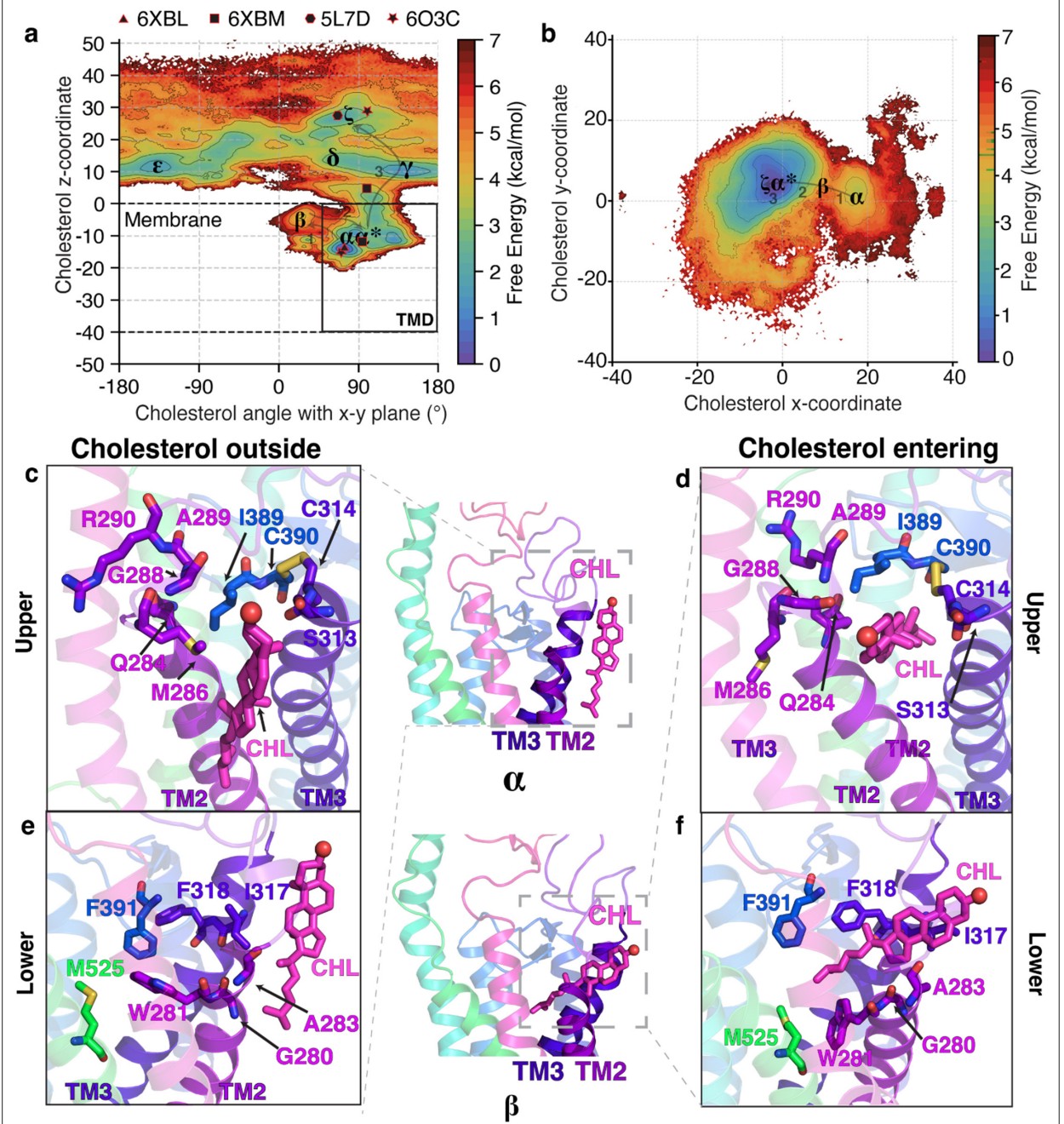

**Figure 2.** The molecular events as cholesterol enters the core of SMO's TMD from the outer leaflet of the membrane. (**a**) Free energy plot showing the angle of cholesterol with the x-y plane, the plane of the membrane, versus the z-coordinate of cholesterol for Pathway 1. The pathway followed by cholesterol is α. The experimental structures of SMO are shown as black polygons. (**b**) Free energy landscape of cholesterol's y-coordinate plotted versus cholesterol's x-coordinate. Cholesterol interacts with residues in TM2-TM3 while entering the core TMD of SMO. (**c–f**) Insets show cholesterol's interactions with residues at the membrane-protein interface for Pathway 1. (**c,e**) show cholesterol outside the protein (α), while (**d, f**) show cholesterol entering the protein (β). All snapshots presented are frames taken from MD simulations.

The online version of this article includes the following figure supplement(s) for figure 2:

**Figure supplement 1.** Error in free energies for *Figure 2* and Figure 4.

**Figure supplement 2.** TICA (time-lagged independent component analysis) plot for SMO cholesterol transport - Pathway 1.

**Figure supplement 3.** Minimum energy path taken by cholesterol for both pathways.

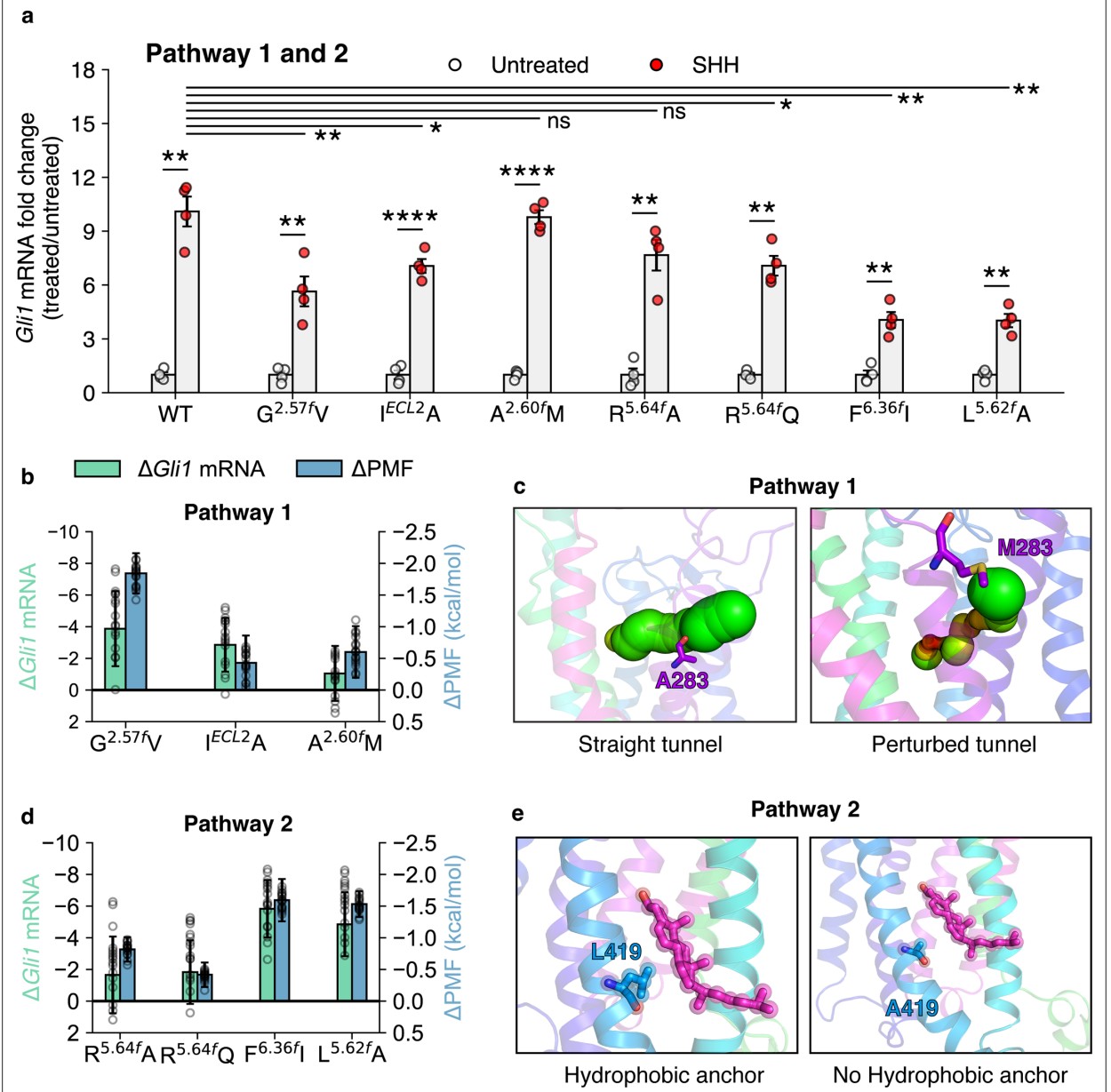

**Figure 3.** Effects of mutations along Pathways 1 and 2 on the activation of SMO. (**a**) *Gli1* mRNA fold changes show the responsiveness of SMO mutants to SHH. Untreated Gli1 levels indicate low SMO activity, while SHH-treated values correspond to the level of SMO activation induced by SHH ligand. A t-test with Welch's correction was used to compute statistical significance. (p values: untreated vs treated: WT: $1.327 \times 10^{-3}$, $G^{2.57f}$ V: $9.212 \times 10^{-3}$, $I^{ECL2}$ A: $4.2 \times 10^{-5}$, $A^{2.60f}$ M: $7.1 \times 100^{-5}$, $R^{5.64f}$ A: $2.062 \times 10^{-3}$, $R^{5.64f}$ Q: $1.192 \times 10^{-3}$, $F^{6.36f}$ I: $2.163 \times 10^{-3}$, $L^{5.62f}$ A: $1.948 \times 10^{-3}$, treated WT vs treated mutant: $G^{2.57f}$ V: $9.1 \times 10^{-3}$, $I^{ECL2}$ A: 0.02734, $A^{2.60f}$ M: 0.7477, $R^{5.64f}$ A: 0.08858, $R^{5.64f}$ Q: 0.02766, $F^{6.36f}$ I: $1.923 \times 10^{-3}$, $L^{5.62f}$ A: $2.306 \times 10$ key: Not significant (ns) p > 0.05, *p ≤ 0.05, **p ≤ 0.01, ***p ≤ 0.001, and ****p ≤ 0.0001, All experimental data represent biological replicates, N=4.) (**b**) *ΔGli1* mRNA fold change (SHH vs untreated) and Δ PMF (difference of peak PMF, calculated as $PMF_{WT} - PMF_{mutant}$) plotted for the mutants in Pathway 1. (**c**) Example mutant $A^{2.60f}$ M shows that cholesterol is able to enter SMO through Pathway 1 even on a bulky mutation. (**d**) Same as (**b**) but for Pathway 2 (**e**) Example mutant $L^{5.62f}$ A shows that cholesterol can enter SMO through Pathway 2 due to lesser steric hindrance. All snapshots presented are frames taken from MD simulations.

The online version of this article includes the following source data and figure supplement(s) for figure 3:

**Figure supplement 1.** Experimental data for mutants along Pathway 1.

**Figure supplement 1—source data 1.** PDF file containing original western blots for *Figure 3—figure supplement 1a*, indicating the relevant bands and treatments.

**Figure supplement 1—source data 2.** Original files for western blot analysis displayed in *Figure 3—figure supplement 1a*.

**Figure supplement 2.** PMF data for mutants along Pathway 1.

*Figure 3 continued*

**Figure supplement 3.** Tunnel diameter profile for WT versus mutant A283M.

**Figure supplement 4.** Experimental data for mutants along Pathway 2.

**Figure supplement 4—source data 1.** PDF file containing original western blots for *Figure 3—figure supplement 4a*, indicating the relevant bands and treatments.

**Figure supplement 4—source data 2.** Original files for western blot analysis displayed in *Figure 3—figure supplement 4a*.

**Figure supplement 5.** PMF data for mutants along Pathway 2.

flexibility of TM2 prevents cholesterol entry into the TMD. Consequently, the activity of mSMO showed a decrease (*Figure 3—figure supplement 1*). However, this decrease could also be attributed to the steric hindrance added by the presence of a bulky propyl group in valine. We designed I$^{ECL2}$A to check the importance of hydrophobic contacts during translocation along Pathway 1. However, this mutant did not affect the activity nor the barrier for translocation significantly (*Figure 3b*, *Figure 3—figure supplement 2*). Finally, we mutated A283$^{2.60f}$ to methionine - A$^{2.60f}$M to test whether the presence of a bulkier residue would block translocation. Surprisingly, the effect on activity was not significant. When we calculated the PMF for cholesterol entry, A$^{2.60f}$M mutant showed a restricted tunnel, but it did not fully block the tunnel (*Figure 3—figure supplement 3*). Therefore, the change in the PMF and experimentally measured activity was not significant (*Figure 3b, c*, *Figure 3—figure supplement 2*).

## Cholesterol 'flipping' corresponds to the highest barrier in its translocation from the inner leaflet along Pathway 2

To quantitatively assess the validity of cholesterol translocation from the inner leaflet, we performed adaptive sampling simulations to obtain the associated free energy barriers. For Pathway 2, cholesterol first binds at the interface between TM5 and TM6 in the inner leaflet (*Deshpande et al., 2019*; *Huang et al., 2018*). In a structure resolved in 2022, cholesterol was observed at the interface between the protein and the membrane, in the inner leaflet, between TM5 and TM6 (*Zhang et al., 2022*). A striking observation is that this cholesterol binding site pose was never used as a starting point for simulations and was discovered independently from the pose described in *Zhang et al., 2022* (*Figure 4—figure supplement 1*). The cholesterol in the inner leaflet has a downward orientation, with the polar hydroxyl group pointing intracellularly ($\eta$, *Figure 4*, *Figure 4—figure supplement 1*). Thus, if cholesterol has to translocate from the inner leaflet, it has to undergo 'flipping' motion, as the resolved structure with the cholesterol completely inside the TMD (*Qi et al., 2020*) shows the alcoholic moiety pointing towards the CRD (*Deshpande et al., 2019*; *Figure 1a*). The energetic barrier associated with flipping the motion of cholesterol can be observed by estimating the angle cholesterol makes with the x-y plane, and the barrier associated with translocation towards the TMD binding site can be estimated by projecting the data on the z-coordinate of cholesterol, similar to *Figure 2a* for Pathway 1.

In *Figure 4a*, multiple free energy minima are observed. The state $\eta$ corresponds to cholesterol outside of the TMD and pointing downwards, forming a –90° angle with the x-y plane. This observed pose is similar to the binding pose for cholesterol outlined in a previously resolved structure, further corroborating our observations (*Zhang et al., 2022*). The state $\epsilon$ corresponds to cholesterol at the membrane-TMD interface and forms a 45° angle with the x-y plane. Finally, the state $\alpha^*$ corresponds to cholesterol inside the TMD and forming a +90° angle with the x-y plane. The state $\alpha^*$, which represents the TMD binding site, is the most stable pose of cholesterol inside SMO. According to the free-energy landscapes (*Figures 2a, 4a* and *Figures 2c, d and 4c, d*), the entry of cholesterol from the inner leaflet is associated with the highest barrier for the entire translocation process, ~6.5 ± 0.8 kcal/mol and it corresponds to the transition between the states $\eta$ and $\theta$. These values are comparable to ATP-Binding Cassette (ABC) transporters of membrane lipids, which use ATP hydrolysis (–7.54 ± 0.3 kcal/mol; *Meurer et al., 2017*) to drive lipid transport from the membrane to an extracellular acceptor. Some of these transporters share the same mechanism as SMO, where the lipid from the inner leaflet is flipped and transported to the extracellular acceptor protein (*Tarling et al., 2013*). Additionally, for secondary active transporters that do not use ATP for the transport of substrates, a thermodynamic barrier of 5–6 kcal/mol has been reported in literature (*Chan et al., 2022*; *Selvam et al., 2019*; *McComas et al., 2023*; *Thangapandian et al., 2025*).

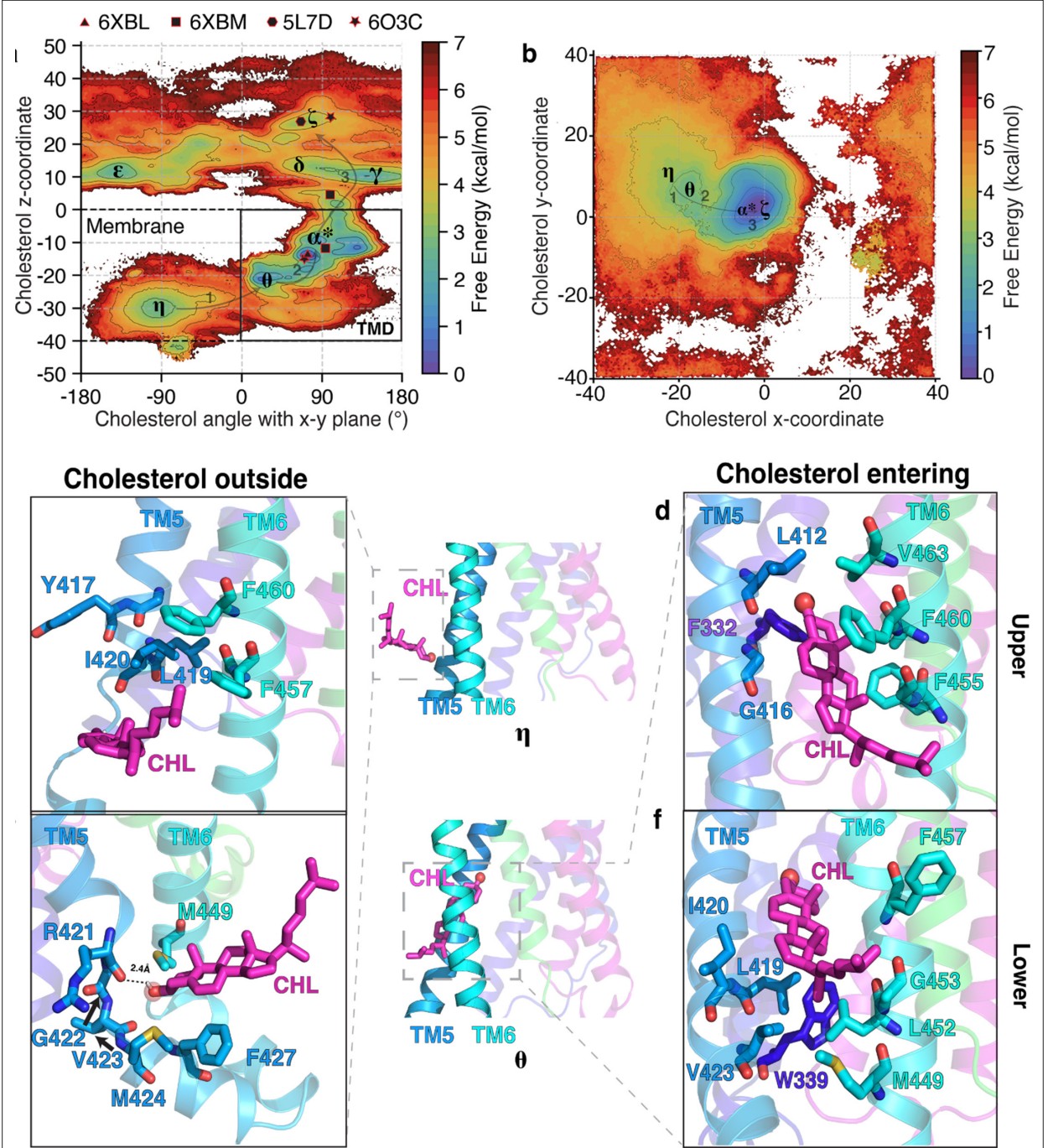

**Figure 4.** The molecular events as cholesterol enters the TMD from the inner leaflet in Pathway 2. (**a**) Free energy plot showing the angle of cholesterol with the x-y plane, the plane of the membrane, versus the z-coordinate of cholesterol for Pathway 2. The pathway followed by cholesterol is η. The experimental structures of SMO are shown as black polygons. (**b**) Free energy landscape of cholesterol's y-coordinate plotted versus cholesterol's x-coordinate for Pathway 2. Cholesterol interacts with residues in TM5-TM6 for Pathway 2 while entering the SMO core TMD. (**e–h**) Insets show cholesterol's interactions with residues at the membrane-protein interface for Pathway 2. (**c, e**) show cholesterol outside the protein (η), while (**d, f**) show cholesterol entering the protein (θ) for Pathway 2. All snapshots presented are frames taken from MD simulations.

The online version of this article includes the following figure supplement(s) for figure 4:

**Figure supplement 1.** Comparison of cholesterol entry with existing resolved structure.

The computed values of free energy barriers are dependent on the projections of the data. Upon projecting the data using the z-coordinate of cholesterol versus the angle between the cholesterol and the x-y plane (*Figure 4a*) and the y-coordinate of cholesterol versus the x-coordinate (*Figure 4b*), the barrier between $\eta$ and $\theta$ is different. On plotting the first two components of tICs (*Figure 2— figure supplement 2a,c*), we observe that the energetic barrier between $\eta$ and $\theta$ is ~6.5 ± 0.8 kcal/ mol. We further validate that the slowest degrees of freedom in the model correspond to the entry of cholesterol from the inner leaflet of the membrane into SMO TMD (*Figure 2—figure supplement 2b,d*).

Contrary to Pathway 1, the entry of cholesterol into the TMD happens via the alcoholic moiety first, which forms a hydrogen bond with the backbone oxygen of R421[5.64f] (*Figure 4e*). Furthermore, the TM5 loses some of its helicity between residues R421[5.64f]-M424[5.67f] due to the flexibility provided by G422[5.65f], which is a part of the conserved WGM motif implicated in SMO activation (*Bansal et al., 2023*). Here, we also designate the 'upper' and 'lower' pocket, which coordinate with the isooctyl and androsterolic moieties of cholesterol, respectively (*Figure 4c–f*). In the upper pocket, the isooctyl tail entry is blocked by the strong hydrophobic patch formed by Y417[5.60f], L419[5.62f], I420[5.63f], F457[6.38f], and F460[6.41f] (*Figure 4c and e*). Flexible residues such as glycine are also not present, which would allow for the fluctuations leading to the opening of the hydrophobic patch and enable entry of the cholesterol tail. Therefore, the androsterolic moiety of cholesterol enters first, followed by the isooctyl tail of cholesterol to reach the state $\epsilon$. To stabilize and facilitate the androsterolic moiety's entry, F460[6.41f] and F455[6.36f] form $\pi - \pi$ contacts, and L419[5.62f] and L452[6.33f] form hydrophobic contacts (*Figure 4d and f*) in state $\theta$. Once cholesterol has flipped in $\theta$, it allows for further translocation towards the TMD binding site ($\alpha^*$) and finally arrival at the CRD site ($\zeta$) with the hydroxyl group pointing extracellularly. The translocation of cholesterol from the membrane to the CRD binding site via Pathway 2 is captured by a $\eta \rightarrow \theta \rightarrow \alpha^* \rightarrow \zeta$ transition. For further clarity, we have plotted the minimum energy path taken by cholesterol as it translocates along this pathway (*Figure 2—figure supplement 3c, d*). Overall, the angle between cholesterol and the x-y plane involves an entire cycle of –90° → 0° → 90° and cholesterol must flip for cholesterol to enter the protein.

Interestingly, mutants along Pathway 2 showed a significant decrease in activity compared to Pathway 1 (*Figure 3a, d*, *Figure 3—figure supplement 4*), along with an increased thermodynamic barrier for translocation (*Figure 3d*, *Figure 3—figure supplement 5*). Mutating R421[5.64f] to alanine or glutamine did not decrease SMO activity significantly (*Figure 3d*), because the interaction with cholesterol is mediated by the protein backbone, and not the side chain (*Figure 4e*). However, mutations like F[6.36f]I and L[5.62f]A reduce SMO activity. Their expression levels of these mutants are comparable to the wild-type mSMO (*Figure 3—figure supplement 4*). The mutants compared to WT SMO showed a significant increase in PMF, due to the lack of the hydrophobic $\pi$-stacking provided by F455[6.36f] and hydrophobic contacts provided by L419[5.62f] during cholesterol translocation. Overall, we report that the mutants for Pathway 2 show a decrease in the activity of SMO and show a strong correlation between the reduction in activity and the barrier for cholesterol translocation (*Figure 3b and d*). These results validate the role of critical residues involved in cholesterol translocation from the inner leaflet as observed in the simulations.

Based on our experimental and computational data, we conclude that cholesterol translocation can happen via either pathway. This is supported on the basis of the following observations: mutations along Pathway 2 affect SMO activity more significantly, and the presence of a direct conduit that connects the inner leaflet to the TMD binding site. In addition, a resolved structure of SMO in the presence of cholesterol (*Zhang et al., 2022*) shows a cholesterol situated at the entry point from the membrane into the protein between TM5 and TM6, in the inner leaflet. However, we also observe that Pathway 1 shows a lower thermodynamic barrier (5.8 ± 0.7 kcal/mol vs. 6.5 ± 0.8 kcal/mol, p = 0.0013). Additionally, PTCH1 controls cholesterol accessibility in the outer leaflet (*Kinnebrew et al., 2021*). This shows that there is a possibility for transport from both leaflets. One possibility that might alter the thermodynamic barriers is native membrane asymmetry, particularly the anionic lipid-rich inner leaflet. This presents as a limitation of our current model.

## The pathway connecting the TMD to the CRD binding sites shows off-pathway intermediate

According to the dual-site model, to reach the binding site in the CRD ($\zeta$), cholesterol translocates along the TMD-CRD interface from the TM binding site ($\alpha^*$) is required. This Common Pathway is shared by cholesterol molecules translocating from the inner and outer leaflets. The translocation of cholesterol from the TMD to the CRD binding site involves a linear movement of the androsterolic moiety through the extracellular end of the TMD, where cholesterol maintains a primarily upright position, with the polar androsterolic moiety pointing towards the CRD site. The energetic barrier associated with the transition ($\alpha^*$ (TMD site) $\rightarrow \zeta$(CRD site)) can be visualized by plotting the z-coordinate of cholesterol versus the angle it forms with the x-y plane (*Figures 2a and 4a*). We observe that the highest barrier along the Common Pathway is ~5.1 ± 0.3 kcal/mol, which is lower than the highest energetic barrier for cholesterol entry from the inner and outer leaflet. Another interesting observation is that the stability of the cholesterol in the TMD binding site ($\alpha^*$) is higher than the CRD binding site ($\zeta$). This observation can be explained on the basis of the thermodynamic driving forces in the two pockets. The TMD binding site is composed mainly of hydrophobic residues, which form strong interactions with cholesterol (*Figure 5—figure supplement 1*) in contrast to the CRD binding site, where the cholesterol is exposed to the aqueous environment. The CRD binding site being exposed

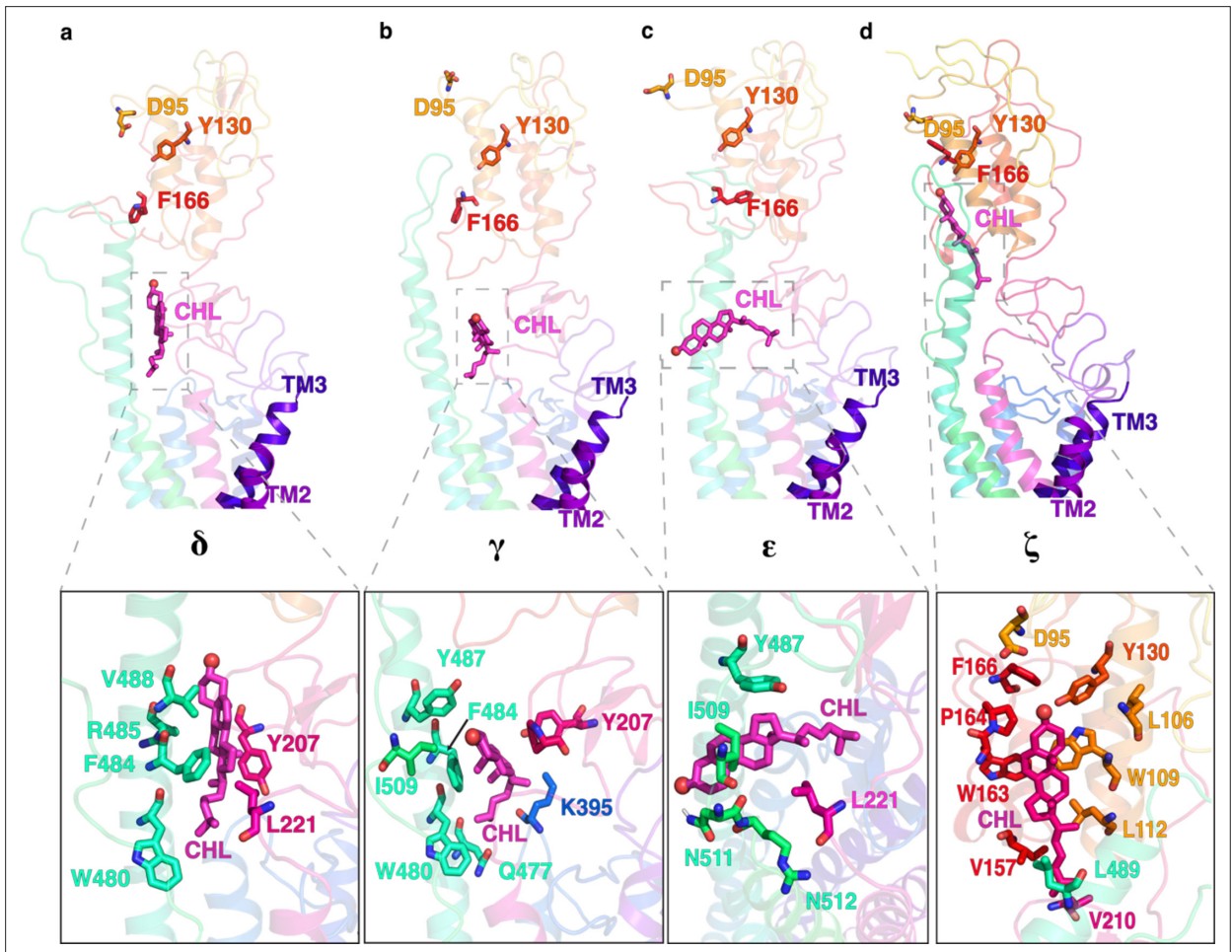

**Figure 5.** Multiple positions of cholesterol as it translocates through the Common Pathway, including the off-pathway intermediate. (**a**) upright ($\delta$), (**b**) tilted ($\gamma$), (**c**) the overtilted off-pathway intermediate ($\epsilon$), and (**d**) cholesterol at the CRD binding site ($\zeta$). All snapshots presented are frames taken from MD simulations.

The online version of this article includes the following figure supplement(s) for figure 5:

**Figure supplement 1.** Hydrophobic tunnel inside SMO core TMD.

**Figure supplement 2.** Cholesterol (purple) forms a tilted pose to enter the CRD binding site.

to the solvent increases the conformational entropy associated with the cholesterol compared to the restrained TMD binding site. In addition, the CRD binding site is more flexible than the TMD binding site in the absence of cholesterol, as reported previously (*Kinnebrew et al., 2022*). This increased flexibility of the CRD binding site further leads to the formation of multiple conformational states between the CRD and cholesterol.

Once cholesterol reaches the TMD-CRD interface, it can adopt multiple poses before reaching the CRD binding site (*Figures 2a and 4a*). Cholesterol at this position can be upright ($\delta$, *Figures 2a and 4a*), where it interacts with F484$^{6.65f}$, W480$^{6.61f}$, V488$^{6.69f}$, and L221$^{LD}$, forming hydrophobic contacts (*Figure 5a*). However, there exists a thermodynamic barrier to take a completely upright path to the CRD binding site (angle ~+90°). This is due to the presence of the long beta sheet of the linker domain (residues L197$^{LD}$-I215$^{LD}$) of SMO at the TMD-CRD interface, blocking the direct upright translocation of cholesterol (*Figure 5—figure supplement 2*). Hence, the major conformation of cholesterol at this position is slightly tilted, away from the plane of the membrane ($\gamma$, *Figures 2a and 4a*). Additionally, in $\gamma$, Y207$^{LD}$ creates a hydrophobic interaction with cholesterol, stabilizing the bent pose (*Figure 5b*).

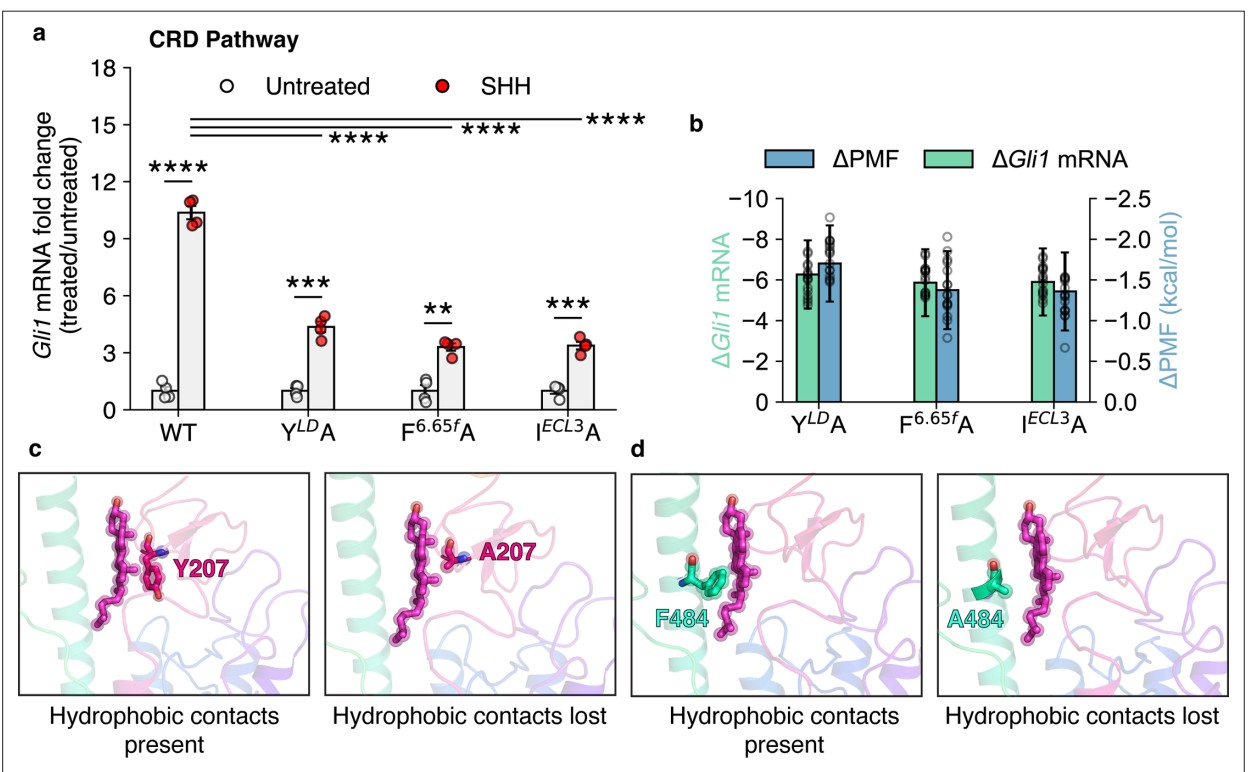

**Figure 6.** The effects of mutations along the translocation pathway connecting the TMD and CRD binding sites on the activation of SMO. (**a**) *Gli1* mRNA fold changes show the responsiveness of SMO mutants to SHH. Untreated Gli1 levels indicate low SMO activity, while SHH-treated values correspond to the level of SMO activation induced by SHH ligand. A t-test with Welch's correction was used to compute statistical significance. (p values: untreated vs treated: WT: $3 \times 10^{-6}$, Y$^{LD}$ A: $2.46 \times 10^{-4}$, F$^{6.65f}$ A: $1.08 \times 10^{-3}$, I$^{ECL3}$ A: $1.12 \times 10^{-4}$, treated WT vs treated mutant: F$^{6.65f}$ A: $1.6 \times 10^{-5}$, I$^{ECL3}$ A: $1.6 \times 10^{-5}$, Y$^{LD}$ A: $1.4 \times 10^{-5}$, key: Not significant (ns) p > 0.05, *p ≤ 0.05, **p ≤ 0.01, ***p ≤ 0.001, and ****p ≤ 0.0001, All experimental data represent biological replicates, N=4.) (**b**) $\Delta$*Gli1* mRNA fold change (SHH vs untreated) and $\Delta$ PMF (difference of peak PMF, calculated as $PMF_{WT}$ - $PMF_{mutant}$) are plotted for mutants along the TMD-CRD pathway. (**c, d**) Example mutants Y$^{LD}$ A and F$^{6.65f}$ A show that cholesterol is unable to translocate through this pathway because of the loss of crucial hydrophobic contacts provided by Y207 and F484 and along the solvent-exposed pathway.

The online version of this article includes the following source data and figure supplement(s) for figure 6:

**Figure supplement 1.** Experimental data for mutants along the Common Pathway between the TMD and the CRD.

**Figure supplement 1—source data 1.** PDF file containing original western blots for *Figure 6—figure supplement 1a*, indicating the relevant bands and treatments.

**Figure supplement 1—source data 2.** Original files for western blot analysis displayed in *Figure 6—figure supplement 1a*.

**Figure supplement 2.** PMF data for mutants along the Common Pathway between the TMD and the CRD.

**Figure supplement 3.** Cholesterol positions along the entire transport pathway from membrane to CRD.

$\gamma$ has been identified as the binding site of the synthetic agonist SAG in SMO (*Qi et al., 2020*). This provides further validation for $\gamma$ being the major intermediate state in the Common Pathway.

Since the degrees of freedom accessible to cholesterol at this point in the pathway are higher than at the TMD, cholesterol can undergo 'overtilting' as it approaches the CRD ($\epsilon$, *Figure 5c*). This state ($\epsilon$) is attributed to an off-pathway intermediate state in the cholesterol translocation process, raising the timescales required for cholesterol to translocate to the CRD binding site. This over-tilted pose is stabilized by hydrophobic interactions between the sterol and Y487$^{6.78f}$, L221$^{LD}$, and I509$^{ECL3}$, and a hydrogen bond between the side chain of N511$^{ECL3}$ and the alcoholic oxygen (*Figure 5c*). Once the cholesterol has crossed the TMD-CRD interface, it can reach the CRD sterol-binding site ($\zeta$, *Figure 5d*) in the CRD. In summary, we identify numerous conformational states of cholesterol-bound SMO, which are distinct from the available structures of sterol-bound SMO.

We performed experimental mutagenesis to validate the critical residues along this pathway, from the TMD to the CRD binding site. These mutants - Y$^{LD}$ A, F$^{6.65f}$ A, and I$^{ECL3}$ A, all showed a significant decrease in activity compared to WT SMO (*Figure 6a, b*, *Figure 6—figure supplement 1*) as well as an increase in peak PMF, suggesting that the force required to translocate cholesterol is higher than the WT residue in this position (*Figure 6b*, *Figure 6—figure supplement 2*). This further implies that the mutants reduce the activity of SMO by increasing the energetic barriers for cholesterol translocation. In particular, the effect is pronounced for Y$^{LD}$ A, which forms hydrophobic contacts along the pathway (*Figure 6c*). F$^{6.65f}$ A showed a significant decrease in SMO activity, which can be attributed to a reduction in hydrophobic stabilizing contacts that enable cholesterol's entry to the CRD (*Figure 6d*).

Overall, the entire cholesterol translocation process can be divided into two parts - entry from the membrane to the TMD binding site via Pathway 1 or 2, and translocation from the SMO TMD to the CRD binding site via the Common Pathway. The entire process is mapped out in *Figure 6—figure supplement 3*. The computational investigation shown here covers the dual-site model, where cholesterol reaches the CRD site via binding to the TM binding site first. In comparison to the CRD site, the TM site is more stable by ~2 kcal/mol (*Figure 2—figure supplement 3b, d*). The experimental and computational analysis shows that both Pathway 1 and Pathway 2 are thermodynamically feasible pathways and exhibit similar energetic barriers for cholesterol to take from the membrane to the TMD binding site.

## A squeezing mechanism for translocation of cholesterol in SMO

To further elucidate the structural changes that happen in SMO during cholesterol translocation, we sought to characterize the conformation of the hydrophobic tunnel inside SMO. We calculated the tunnel diameter along the channel as cholesterol traverses through the protein (*Figure 7* and *Figure 7—figure supplement 1*). The tunnel calculations were done with cholesterol at different points along the pathway as indicated by the metastable states in the Markov state models - with cholesterol in the TMD binding site, cholesterol at the CRD-TMD interface, and cholesterol present at the binding site in the CRD. When we plot cholesterol's position at different points along its transit through the TMD, we observe that the tunnel diameter varies along the z-coordinate as cholesterol moves through the channel. This corroborates that the tunnel radius is a function of the position of cholesterol in SMO. Furthermore, we observe that the tunnel radius shows a peak at z ~ -7 when cholesterol is present in the core of the TMD (*Figure 7a, b*). In the rest of the tunnel, the average radius remains relatively small (*Figure 7—figure supplement 2*). This peak in diameter is seen to move along with the cholesterol position (*Figure 7c, d*). This provides evidence that SMO uses a squeezing mechanism to translocate cholesterol. The term 'squeezing' here implies that the tunnel remains open only around cholesterol, and it closes as cholesterol moves away from its current position in the tunnel. Additionally, recent experimental work (*Zhang et al., 2022*) also suggests a squeeze-type mechanism for SMO. Once cholesterol has reached the binding site in the CRD, the tunnel in the TMD region is closed (*Figure 7e, f*, *Figure 7—figure supplement 2*).

The cholesterol translocation mechanism of SMO shows similarities to the alternating access model proposed for substrate transport in membrane transporter proteins, where the transport tunnel closes behind the substrate to facilitate substrate transport across the membrane. Membrane transporters, including lipid transporters, use either an ion-based gradient or ATP hydrolysis to facilitate substrate transport. An ion binding site for SMO is currently unknown in contrast to Class A receptors, which have a sodium binding site (*Katritch et al., 2014*; *Selvam et al., 2018*; *Dutta et al., 2022b*). There

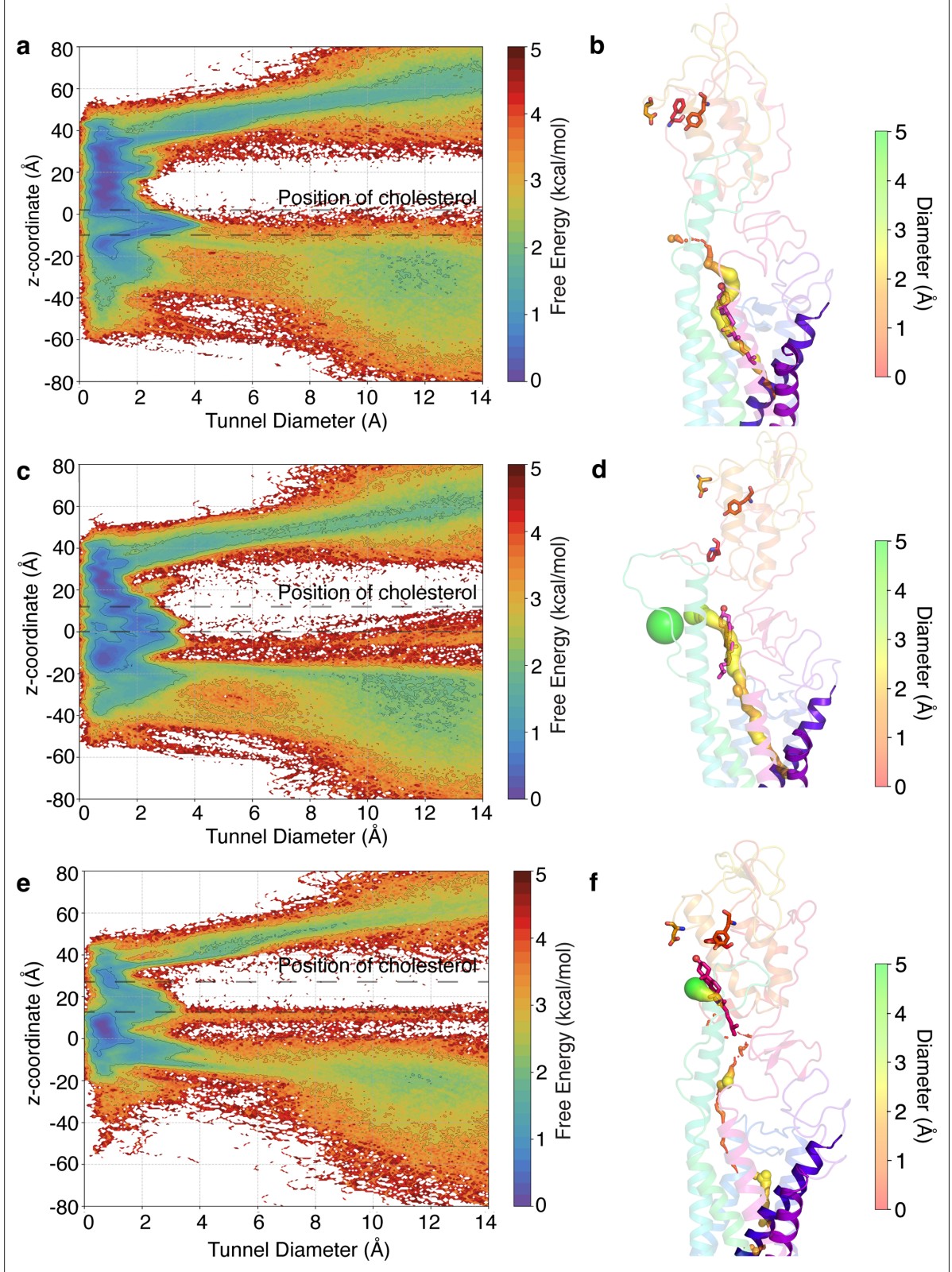

**Figure 7.** The tunnel profile during cholesterol translocation in SMO. (**a**) Free energy plot of the z-coordinate versus the tunnel diameter when cholesterol is present in the core TMD. The tunnel shows a spike in the radius in the TMD domain, indicating the presence of a cholesterol-accommodating cavity. (**b**) Representative figure for the tunnel when a cholesterol molecule is in the TMD. (**c**) Same as (**a**), when cholesterol is at the TMD-CRD interface. (**d**) same as (**b**), when cholesterol is at the TMD-CRD interface. (**e**) same as (**a**), when cholesterol is at the CRD binding site. (**f**) same

*Figure 7 continued on next page*

*Figure 7 continued*

as (**b**), when cholesterol is at the CRD binding site. Tunnel diameters are shown as spheres. Cholesterol positions are marked on plots using dotted lines. All snapshots presented are frames taken from MD simulations.

The online version of this article includes the following figure supplement(s) for figure 7:

**Figure supplement 1.** Error in Free Energies for *Figure 7*.

**Figure supplement 2.** Average tunnel radius for *Figure 7*.

is no experimental evidence of ion-coupling with the cholesterol export via SMO. Therefore, we posit that cholesterol translocation through SMO involves passive or concentration-dependent diffusion driven by a shift in the pool of accessible cholesterol, which rises once PTCH1 is inhibited. Thus, we provide a mechanistic overview of the dynamics of the cavity during cholesterol translocation in SMO.

## The translocation of cholesterol occurs on a millisecond timescale

To give a perspective on the overall translocation process from a kinetic standpoint, we sought to calculate the timescales associated with the translocation of cholesterol from the membrane to the binding site in the CRD. Using a combination of Transition Path Theory and Markov state models, the reactive fluxes associated with each stage of the translocation cycle can be calculated. This enables us to calculate the mean first passage time (MFPT), which gives us an estimate of the timescales associated with the process (*Figure 8*). Pathway 1 was divided into two stages - the first being the translocation of cholesterol from the outer leaflet (*Figure 8d*) to the TMD - yielding a mean first passage time of 700 ± 122 $\mu$s. Here, the cholesterol tail has entered the TMD (*Figure 8a*). This was followed by cholesterol reaching the TMD binding site, with a timescale of 205 ± 41 $\mu$s (*Figure 8b*). For Pathway 2, cholesterol starts in the inner leaflet (*Figure 8f*). In the first stage, cholesterol undergoes flipping, which leads to a higher timescale than stage 1 of Pathway 1–823 ± 132 $\mu$s. This is followed by 122 ± 22 $\mu$s for cholesterol to reach the TMD binding site (*Figure 8b*).

Once cholesterol has reached the TMD binding site, it is followed by translocation of cholesterol from the TMD to the CRD binding site with a timescale of 255 ± 36 $\mu$s. Overall, the calculated timescales for cholesterol to reach the CRD site from the inner/outer leaflet of the membrane are 1023 ± 223 $\mu$s (Pathway 1) and 1134 ± 188 $\mu$s (Pathway 2). These timescales are comparable to the substrate transport timescales of Major Facilitator Superfamily (MFS) transporters (*Chan et al., 2022*). Furthermore, several experimental studies have also resolved the millisecond-scale kinetics of MFS transporters (*Blodgett and Carruthers, 2005*; *Körner et al., 2024*; *Bazzone et al., 2022*; *Smirnova et al., 2014*; *Zhu et al., 2019*), further corroborating the results from our study. Interestingly, the timescales for the reverse process of translocating cholesterol from the CRD binding site to the membrane are higher for each step (*Figure 8*), indicating that the reverse process has a higher energetic barrier associated with it. This provides kinetic evidence that the overall translocation process from the membrane to the CRD binding site is thermodynamically favorable. Thus, it can be concluded that SMO facilitates the translocation of cholesterol.

## Conclusion

In this study, we have used a combination of millisecond-scale atomistic molecular dynamics simulations, Markov state modeling, and experimental mutagenesis to describe the step-by-step process of cholesterol translocation through SMO. Previous structural studies have delineated multiple translocation pathways for cholesterol transport via SMO. In this study, we have examined the mechanism of cholesterol translocation from the membrane to the CRD sterol-binding site of SMO via two modes - the outer leaflet pathway, between TM2 and TM3, and the inner leaflet pathway, between TM5 and TM6. We quantitatively assess the thermodynamic barriers of the translocation pathways and estimate the timescales associated with the process. The key intermediate cholesterol-bound conformations of SMO and the role of specific residues in the translocation process were identified computationally and validated using both experimental and in silico mutagenesis.

We observe that cholesterol can move through two distinct conduits, starting at either the outer leaflet of the membrane (Pathway 1) or the inner leaflet (Pathway 2), followed by translocation to the cholesterol binding site in the CRD. In the first mode (Pathway 1), cholesterol enters the protein between TM2 and TM3 in the outer leaflet, and then it is translocated along the extended TM6 to

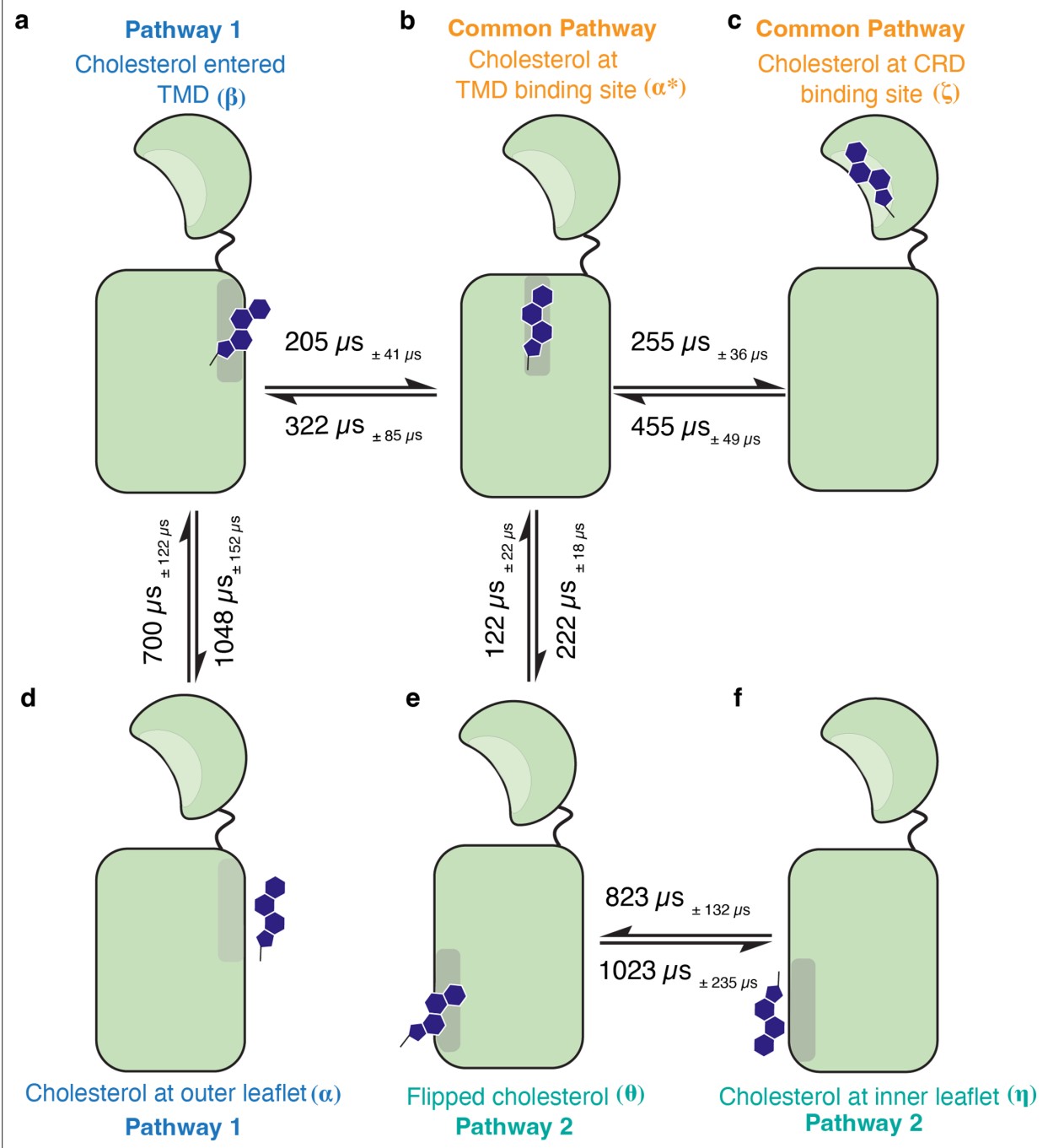

**Figure 8.** The timescales associated with the translocation of cholesterol through SMO. Each major intermediate state has been marked (**a–f**). Timescales were obtained by calculating the mean first passage time (MFPT) using the Markov state model. Errors in timescales are shown as subscripts. The arrows represent the relative flux for the translocation between subsequent steps. The overall process occurs at a timescale of ~1 ms.

reach the CRD binding site. The highest barrier along this first mode is associated with the cholesterol entry from the outer leaflet of the membrane to the TMD binding site. Similarly, the highest barrier for the second mode (Pathway 2) also involves cholesterol translocation from the inner leaflet membrane to the TMD of SMO. We show that the barriers associated with the pathway starting from the outer leaflet are lower by ~0.7 kcal (p=0.0013). We also provide evidence that cholesterol can enter SMO via both leaflets, considering that multiple computational and experimental studies have found cholesterol entry sites and activation modulation via the outer leaflet, between TM2-TM3 (*Hedger et al., 2019*; *Kinnebrew et al., 2021*; *Bansal et al., 2023*). Other experimental and computational studies

have proposed that cholesterol can enter SMO via the inner leaflet (*Huang et al., 2018*; *Zhang et al., 2022*). Overall, our work shows that cholesterol translocation from either pathway is feasible.

The second-highest barrier in cholesterol translocation is at the TMD-CRD interface, but it is ~1 kcal/mol lower than the barrier for cholesterol entry. We also show that the cholesterol translocation process occurs via a squeezing mechanism that maintains the forward flux of cholesterol from the membrane to the CRD binding site. Overall, the translocation process takes place on the millisecond timescale with multiple intermediate states identified using simulations and supported by the sterol-bound structures of SMO.

Despite the extensive MD simulations reported in this study, there is still a need to further probe the endogenous "activation" of SMO by cholesterol in a position-dependent manner. Here, we have explored the role the CRD site plays in SMO activation. In addition, through simulating the CRD site-dependent SMO activation hypothesis, we have also simulated the TMD site-dependent activation. We show that the overall stability of cholesterol in the TMD site is higher than the stability of cholesterol in the CRD site by ~2 kcal/mol. An alternative possibility states that the flexibility associated with the CRD would allow it to directly access the membrane and, consequently, cholesterol. In the extensive simulations reported in this study, the binding site of cholesterol in the CRD remains at least 20 Å away from the nearest lipid head group in the membrane, suggesting that such direct extraction and the bending of the CRD do not occur within the timescales sampled (*Appendix 2—figure 6*). The mechanistic details of this process are still unexplored and form the basis of future work. Additionally, GPCRs exist in a conformational equilibrium of the active and inactive states, and the job of the agonist (cholesterol) is to lower the barrier to activation and shift the equilibrium towards the active state. Our recent work (*Kim et al., 2024*) discusses how the binding position of cyclopamine modulates SMO activity. Cyclopamine acts as an antagonist when bound to SMO TMD and acts as an agonist when bound to SMO CRD. Cholesterol binding at different positions along the translocation pathway leads to a position-dependent modulation of SMO activity (*Kinnebrew et al., 2022*; *Kumari et al., 2023*). Kinnebrew et al. propose that binding of cholesterol to the TMD determines basal SMO activity, whereas binding to both the TMD and CRD is required for full, SHH-induced activity. In this study, we only focused on the cholesterol movement from the membrane to the CRD binding site. Therefore, a future investigation is needed to fully sample the activation process of SMO with cholesterol bound at different positions.

Our study provides computational and experimental evidence for the translocation of cholesterol along the conduit within SMO and outlines the entire translocation mechanism from a kinetic and thermodynamic perspective. Our findings provide a plausible model for how an increase in membrane cholesterol accessibility can activate SMO: by promoting the entry and movement of cholesterol along a conduit that traverses the center of the protein and ends at the CRD. Our study provides a framework for the development of drugs against oncogenic SMO that could function by blocking intermediate states of transport.

## Methods

**Key resources table**

| Reagent type (species) or resource | Designation | Source or reference | Identifiers | Additional information |
|---|---|---|---|---|
| Cell line (*Mus musculus*) | Smo⁻/⁻ MEFs | *Rohatgi et al., 2007* | – | Mouse embryonic fibroblasts lacking Smoothened |
| Antibody | Anti-GLI1 (Mouse monoclonal, clone L42B10) | Cell Signaling Technology | Cat# 2643; RRID:AB_2294746 | Western Blot, 1:1000 dilution |
| Antibody | Anti-SMO (Rabbit polyclonal) | *Rohatgi et al., 2007* | RRID:AB_3738384 | Western Blot, 1:2000 dilution |
| Antibody | Anti-GAPDH (Mouse monoclonal, clone 1E6D9) | ProteinTech | Cat# 60004–1-Ig; RRID:AB_2107436 | Western Blot, 1:10,000 dilution |
| Chemical compound, drug | High-glucose DMEM | Thermo Fisher Scientific | Cat# SH30081FS | – |
| Chemical compound, drug | Fetal Bovine Serum (FBS) | Sigma-Aldrich | Cat# S11150 | – |

*Continued on next page*

*Continued*

| Reagent type (species) or resource | Designation | Source or reference | Identifiers | Additional information |
|---|---|---|---|---|
| Chemical compound, drug | Sodium pyruvate | Gibco | Cat# 11-360-070 | – |
| Chemical compound, drug | L-glutamine | GeminiBio | Cat# 400106 | – |
| Chemical compound, drug | Minimum essential medium NEAA solution | Gibco | Cat# 11140076 | – |
| Chemical compound, drug | Penicillin / Streptomycin | GeminiBio | Cat# 400109 | – |
| Chemical compound, drug | SigmaFast Protease inhibitor cocktail, EDTA-free | Sigma-Aldrich | Cat# S8830 | – |

## Molecular Dynamics (MD) simulations

### Simulation setup

Structures of SMO bound to sterols - 6XBL (*Qi et al., 2020*; SMO-CHL-1), 6XBM (*Qi et al., 2020*; SMO-CHL-2, SMO-CHL-3), and 5L7D (*Byrne et al., 2016*; SMO-CHL-4) were used as the starting structures for simulations. For SMO-CHL-1, the bound agonist SAG was removed. The two sterols occupying different positions in the tunnel in 6XBM were used to build 2 separate systems with cholesterol at different sites in the pathway (SMO-CHL-2, SMO-CHL-3). For SMO-CHL-2 and SMO-CHL-3, to account for the lack of CRD in the structure 6XBM, the sterol positions were aligned to the full-length SMO (6XBL, 0.8 Å RMSD from 6XBM) and the 24(S),25-epoxycholesterol for each system was replaced by cholesterol. For SMO-CHL-4, the inactivating mutation $V^{3.40}F$ was mutated back to Wild Type. For all systems, any stabilizing antibodies and bound G proteins were removed. The missing residues in the intra/extracellular loop for every protein were modeled using MODELLER (*Webb and Sali, 2016*; *Appendix 1—table 1*). Termini for all proteins were capped using acetyl (ACE) and N-methyl-amino (NME) at the N- and C-termini to ensure neutrality. All four protein systems with cholesterols at different points along the translocation path were embedded in a lipid bilayer. The composition of the bilayer was set similar to mice cerebellum (*Scandroglio et al., 2008*; *Appendix 1—table 2*), to mimic physiological conditions, using CHARMM-GUI (*Jo et al., 2008*; *Lee et al., 2019*). Interactions between the atoms - bonded and non-bonded - were modeled using the CHARMM36 Force Field (*Klauda et al., 2010*; *Best et al., 2012*). TIP3P water (*Jorgensen et al., 1983*) and 0.15 M NaCl were used to solvate the system, to mimic physiological conditions. Non-protein hydrogen masses were repartitioned to 3.024 Da, to enable use of a longer timestep (4 fs; *Hopkins et al., 2015*). Starting points for Pathways 1 and 2 were chosen from already simulated data, according to the closest cholesterol distance from the respective helices (outer leaflet, TM2-TM3 for Pathway 1, lower leaflet, TM5-TM6 for Pathway 2). This was done once the rest of the pathway was completely explored.

### Pre-production MD

All systems were subject to 50,000 steps of initial minimization. Further, minimization was performed for another 10000 steps by constraining the hydrogens using SHAKE (*Andersen, 1983*). Systems were then heated to 310 K to mimic physiological conditions, at NVT for 10 ns. This was followed by equilibration at NPT and 1 bar for 5 ns. Backbone constraints of 10 kcal/mol/Å$^2$ were applied during NVT and NPT. Next, systems were equilibrated at NPT for 40 ns without constraints to ensure system stability. All pre-production steps were performed using AMBER18 (*Case et al., 2020*; *Salomon-Ferrer et al., 2013b*; *Case et al., 2005*; *Götz et al., 2012*; *Salomon-Ferrer et al., 2013a*).

### Production MD

Systems were then subjected to some initial sampling of 100 ns each. This was followed by clustering and performing adaptive sampling to enable a divide-and-conquer approach to sample the conformational landscape. Three rounds of sampling were performed on each system. This was followed by similar rounds of adaptive sampling on the distributed computing project Folding@Home (http://foldingathome.org). OpenMM 7.5.1 (*Eastman et al., 2017*) was used for running simulations on Folding@Home.

In all simulations, 4 fs was the chosen integration timestep. Particle Mesh Ewald (PME; *Darden et al., 1993*) method was used to account for long-range electrostatics. The cutoff for considering non-bonded interactions was set to 10 Å. Temperature was maintained using the Langevin Thermostat (*Davidchack et al., 2009*). Pressure was maintained using the Monte Carlo Barostat (*Åqvist et al., 2004*). All hydrogen bonds were constrained using SHAKE (*Andersen, 1983*).

## Steered MD

Steered MD was performed to steer cholesterol from the membrane into the protein, for generating the starting frames for Pathway 1 and Pathway 2. This was done by first finding frames from the existing data, where cholesterols were closest to the respective starting points. Then, cholesterols were steered towards the center of TMD with the end point being the cholesterol binding site deep in the TMD, as resolved in the structure 6XBL. This was done using a steering force of 20 kcal/(mol Å²), over a course of 500 ns. The entire protein, except the helices involved at the entry (TM2-TM3: Pathway 1; TM5-TM6: Pathway 2), was constrained using two RMSD restraints during simulations to prevent any unphysical effects. The force constants used for RMSD restraints are as follows: 10 kcal/(mol Å²) for restraining residues from CRD to ICL1, and 35 kcal/(mol Å²) for restraining residues from ICL2 to helix 8 for Pathway 1. For Pathway 2, an RMSD restraint of 35 kcal/(mol e the effect of mutations on the choleste) was used to restrain residues from CRD to ECL2, and 10 kcal/(mol Å²) to restrain residues from ECL3 to helix8. Three replicates of steered MD were performed to ensure that the pathway explored converged. The frames generated from these runs were then used as seed frames to start simulations for exploring the entry of cholesterol into SMO. Steered MD was performed using NAMD (*Phillips et al., 2020*; *Phillips et al., 2005*). Each frame generated by Steered MD was minimized for 50000 steps, and then equilibrated for 40 ns, using AMBER using the same methodology for these seed frames as described in section Pre-Production MD.

## Adaptive biasing force-based sampling for PMF generation

To elucidate the effect of mutations on the cholesterol translocation barriers, we used an adaptive-biasing force (ABF)-based sampling for generating the potential of mean force (PMF) profiles for each case and compared them to the WT translocation barriers. For generating the starting files for every mutant system, psfgen (*Humphrey et al., 1996*; *Stone, 1998*) was used. SMO WT PMF profiles were generated for each pathway, and mutant profiles were generated for their respective pathways. A biasing potential of 45 kcal/(mol Å²) was used for both lower wall and upper wall of the ABF potential. Each mutant was run for 3 replicates, and $10^7$ samples were generated for each pathway to compute the PMF. NAMD was used for this purpose (*Phillips et al., 2020*; *Phillips et al., 2005*).

## Adaptive sampling, feature selection, and clustering

The cholesterol translocation process in SMO was simulated in a stage-wise process, with the four starting points having cholesterol present at different points along the translocation pathway (*Figure 1*). To accelerate the sampling of the entire translocation process, simulations were performed using an approach that parallelizes the exploration of conformational space. Adaptive sampling (*Hinrichs and Pande, 2007*; *Bowman et al., 2010*; *Kleiman et al., 2023*) was utilized to achieve this acceleration, which uses an iterative sampling approach involving picking the next round of simulation starting points based on the current data. Several different types of machine learning and heuristic-based adaptive sampling approaches have been proposed in the literature (*Kleiman et al., 2023*; *Kleiman and Shukla, 2023*; *Kleiman and Shukla, 2022*; *Shamsi et al., 2018*; *Shamsi et al., 2017*; *Zimmerman and Bowman, 2015*; *Weber and Pande, 2011*). However, least count-based sampling, where the least visited states are chosen as the starting points for the next set of simulations, is among the best sampling strategies for exploration of the conformational free energy landscapes (*Weber and Pande, 2011*; *Nadeem and Shukla, 2025*). The following approach was used to collect the data:

1. Initially, sampling was started from the four starting points with cholesterol at different points along the tunnel (*Figure 1*). Each starting point was simulated as a separate system.
2. All systems were subject to Pre-Production MD (Refer Pre-Production MD).
3. Following Pre-Production, all systems were subject to 200 ns of Production MD (Refer Production MD).

4. The production data until this point was combined and clustered on the basis of Adaptive Sampling metrics (*Appendix 1—table 3*) using k-means clustering (for the first four rounds) and then mini-batch k-means later on. pyEMMA (*Scherer et al., 2015*) was used for this purpose.
5. Frames were chosen from the clusters with the least populations and were used as seeds for the next round of simulations. This enabled a parallel iterative approach for sampling the conformational landscape (*Appendix 1—table 4*, *Appendix 2—figure 1*)
6. Steps 4–5 were repeated until the entire landscape was explored.

The entire process of cholesterol translocation - from cholesterol present in the membrane to cholesterol bound to the CRD sterol-binding site - was simulated. A total of 2 ms of unbiased simulation data was collected using this approach.

## Dimensionality reduction using tICA

Dimensionality reduction was performed on the high-dimensional data set using Time-Lagged Independent Component Analysis (tICA) (*Schwantes and Pande, 2013*). tICA uses a linear transformation to project the input data into a lower-dimensional basis set, the components of which approximate the slowest processes observed in the simulations. In our case, the slowest process simulated was identified by tICA as the translocation of cholesterol (*Figure 2—figure supplement 2*). The dataset was divided into 2 different groups, separately for Pathways 1 and 2. To further gain insights from the simulations, we constructed a Markov state model from the tICA-projected simulation data.

## Markov State model construction

Markov State Model (MSM) is a kinetic modeling technique that uses short trajectory data that sample local transitions to provide a global estimate of the thermodynamics and kinetics of the physical process (*Pande et al., 2010*; *Noé and Rosta, 2019*; *Prinz et al., 2011*; *Husic and Pande, 2018*; *Wang et al., 2018*; *Shukla et al., 2015*). A MSM discretizes the dataset into kinetically distinct microstates and calculates the rates of transitions among such microstates. This methodology has been used extensively to investigate the conformational dynamics of membrane proteins, including GPCRs (*Deganutti et al., 2024*; *Fleetwood et al., 2021*; *Dutta et al., 2022a*; *Dutta et al., 2022b*; *Chan et al., 2022*; *Kohlhoff et al., 2014*; *Kim et al., 2024*; *Bansal et al., 2023*). To construct the Markov state model, the data collected from simulations were first featurized (*Appendix 1—table 3*). Pathways 1 and 2 were treated independently of each other. tICA was performed on the data to reduce the dimensionality of the input data and identify the slowest processes observed in the simulations. A Markov state model was then constructed on the 42 and 28 components from the tIC space for Pathways 1 and 2, respectively. To construct the models for both pathways, the following approach was used:

1. The data was clustered into different numbers of clusters, ranging from 100 to 1000, and the implied timescales were calculated as a function of the MSM lag time. The lag time after which the implied timescales converged (30 ns) in both cases was chosen as the MSM lag time (*Appendix 2—figure 2a*, *Appendix 2—figure 3a*).
2. Once the MSM lag times were chosen, a grid-search-based approach was chosen to compute the optimal number of clusters. Each constructed MSM was evaluated using a VAMP2 score (*Wu and Noé, 2020*), where the sum of squares of the top five eigenvalues was computed. Each pathway's dataset was clustered into several clusters (200 - Pathway 1, 400 - Pathway 2) that gave the highest VAMP2 score (*Appendix 2—figure 2b*, *Appendix 2—figure 3b*).
3. Once the MSMs were constructed, bootstrapping was performed, with 200 rounds and 80% of the data in each round, to compute the error associated with the probabilities.
4. To validate the final MSM, a Chapman-Kolmogorov test was performed (*Appendix 2—figures 4 and 5*) to show the long-timescale validity of the Markovian property followed by the constructed models.

This information can be used to calculate the probability of each state, which can be used to recover the thermodynamic and kinetic properties of the entire ensemble. One of the use cases of the probabilities is that they can be used to reweigh the projected free energy of each datapoint along a said reaction coordinate, which has been used in *Figures 2, 4 and 7*. The errors in free energies were computed using the errors in the projected probabilities from the bootstrapped MSMs (*Figure 2—figure supplement 1*, and *Figure 7—figure supplement 1*).

## Trajectory analysis and visualization

Trajectories were stripped of water and imaged before analysis to allow faster computation. cpptraj (*Roe and Cheatham, 2013*) was used for this purpose. For constructing the figures, VMD (*Humphrey et al., 1996*; *Stone, 1998*) and open-source PyMOL (*Schroedinger, 2022*) (rendering), MDTraj (*McGibbon et al., 2015*) (computing observables from trajectories), matplotlib (*Hunter, 2007*) and seaborn (*Waskom, 2021*) (Plot rendering), Numpy (*Harris et al., 2020*) (numerical computations), HOLE (*Smart et al., 1993*) (tunnel diameter calculations) were used.

## Cell culture and cell line generation

Mouse embryonic fibroblasts (MEFs) lacking Smoothened (*Smo*$^{-/-}$) were tested to ensure lack of endogenous SMO protein using immunoblotting, as described previously (*Rohatgi et al., 2007*). These *Smo*$^{-/-}$ MEF cells were used to generate stable cell lines expressing SMO mutants, which were then authenticated by immunoblotting to ensure stable expression of the transgene (*Nachtergaele et al., 2013*). Cell lines were confirmed to be negative for mycoplasma infection.

MEF cells were grown in high-glucose DMEM (Thermo Fisher Scientific, catalog no. SH30081FS) containing 10% FBS (Sigma-Aldrich, catalog no. S11150) and the following supplements: 1 mM sodium pyruvate (Gibco, catalog no. 11-360-070), 2 mM L-glutamine (GeminiBio, catalog no. 400106), 1× minimum essential medium NEAA solution (Gibco, catalog no. 11140076), penicillin (40 U/ml), and streptomycin (40 µg/ml) (GeminiBio, catalog no. 400109). This media, hereafter referred to as supplemented DMEM, was sterilized through a 0.2-µm filter and stored at 4°C.

To measure Hedgehog responsiveness by quantitative polymerase chain reaction (PCR) or western blotting, cells were seeded in 10% FBS-supplemented DMEM and grown to confluence. To induce ciliation, a requirement for Hedgehog signaling, cells were serum-starved in 0.5% FBS-supplemented DMEM and simultaneously treated with Sonic Hedgehog (SHH) for 24 hr before analysis.

Western blotting was carried out to assess SMO protein expression for all mutants. Briefly, whole-cell extracts were prepared in lysis buffer containing 150 mM NaCl, 50 mM Tris-HCl (pH 8), 1% NP-40, 1× protease inhibitor (SigmaFast Protease inhibitor cocktail, EDTA-free; Sigma-Aldrich, catalog no. S8830), 1 mM MgCl$_2$, and 10% glycerol. After lysate clarification by centrifugation at 20,000 × *g*, samples were resuspended in 50 mM tris(2-carboxyethyl)phosphine and 1× Laemmli buffer for 30 min at 37°C. Samples were then subjected to SDS-polyacrylamide gel electrophoresis, followed by immunoblotting with antibodies against GLI1 [anti-GLI1 mouse monoclonal (clone L42B10); Cell Signaling Technology, catalog no. 2643, RRID:AB_2294746], SMO (rabbit polyclonal) (*Rohatgi et al., 2007*), or GAPDH [anti-GAPDH mouse monoclonal (clone 1E6D9); Protein tech, catalog no. 60004–1-Ig, RRID:AB_2107436].

## Measuring Hedgehog signaling with quantitative PCR

*Gli1* mRNA transcript levels were measured using the Power SYBR Green Cells-to-CT kit (Thermo Fisher Scientific). *Gli1* levels relative to *Gapdh* were calculated using the Delta-Ct method CT(*Gli1*) - CT(*Gapdh*). The RT-PCR was carried out using custom primers for *Gli1* (forward primer: 5′-ccaagcca actttatgtcaggg-3′ and reverse primer: 5′-agcccgcttctttgttaatttga-3′), and *Gapdh* (forward primer: 5′-agtggcaaagtggagatt-3′ and reverse primer: 5′-gtggagtcatactggaaca-3′).

## Acknowledgements

The authors thank The Blue Waters Petascale Computing Facility and National Center for Supercomputing Applications, which is supported by the National Science Foundation (awards OCI-0725070 and ACI-1238993) and the state of Illinois. Blue Waters is a joint effort of the University of Illinois at Urbana-Champaign and its National Center for Supercomputing Applications. DS acknowledges support from NIH grant R35GM142745 and the Cancer Center at Illinois. RR acknowledges support from NIH grant GM118082. The authors also acknowledge support from citizen scientists providing computing hours for simulations performed on Folding@Home, which have enabled us to collect data on a large scale for this project.

## Additional information

### Funding

| Funder | Grant reference number | Author |
|---|---|---|
| National Science Foundation | OCI-0725070 | Prateek D Bansal<br>Diwakar Shukla |
| National Science Foundation | ACI-1238993 | Prateek D Bansal<br>Diwakar Shukla |
| National Institutes of Health | R35GM142745 | Prateek D Bansal<br>Diwakar Shukla |
| National Institutes of Health | GM118082 | Maia Kinnebrew<br>Rajat Rohatgi |

The funders had no role in study design, data collection and interpretation, or the decision to submit the work for publication.

### Author contributions

Prateek D Bansal, Conceptualization, Resources, Data curation, Software, Formal analysis, Investigation, Visualization, Methodology, Writing – original draft, Writing – review and editing; Maia Kinnebrew, Validation, Writing – review and editing; Rajat Rohatgi, Resources, Supervision, Funding acquisition, Validation, Writing – review and editing; Diwakar Shukla, Conceptualization, Resources, Supervision, Funding acquisition, Validation, Investigation, Methodology, Writing – original draft, Project administration, Writing – review and editing

### Author ORCIDs

Prateek D Bansal ⓘ https://orcid.org/0000-0002-1887-2320
Maia Kinnebrew ⓘ https://orcid.org/0000-0002-7344-8231
Rajat Rohatgi ⓘ https://orcid.org/0000-0001-7609-8858
Diwakar Shukla ⓘ https://orcid.org/0000-0003-4079-5381

Reviewer #1 (Public review): https://doi.org/10.7554/eLife.108030.3.sa1
Reviewer #2 (Public review): https://doi.org/10.7554/eLife.108030.3.sa2
Reviewer #3 (Public review): https://doi.org/10.7554/eLife.108030.3.sa3
Author response https://doi.org/10.7554/eLife.108030.3.sa4

## Additional files

### Supplementary files

MDAR checklist

### Data availability

Python scripts used for analysis are available on GitHub (https://github.com/ShuklaGroup/Bansal_et_al_Cholesterol_Smoothened_2024, copy archived at *Bansal, 2024*). Relevant .npy files and .pdb files from each microstate of the MSM, used for the generation of plots, have been submitted to a Dryad Repository (https://doi.org/10.5061/dryad.76hdr7t4w). Due to the large size (>2 TB) of the molecular dynamics simulation trajectories generated for this study, they are available by contacting the corresponding author upon reasonable request.

The following dataset was generated:

| Author(s) | Year | Dataset title | Dataset URL | Database and Identifier |
|---|---|---|---|---|
| Bansal P, Kinnebrew M, Rohatgi R, Shukla D | 2025 | Multiple modes of cholesterol translocation in the human Smoothened receptor | https://doi.org/10.5061/dryad.76hdr7t4w | Dryad Digital Repository, 10.5061/dryad.76hdr7t4w |

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

# Appendix 1

## Modeled residues in SMO

**Appendix 1—table 1.** Modeled residues in 5L7D-inactive-Apo-SMO starting structure.
The helical content of K440–I445 was modeled based on the structure of SANT1-bound SMO (PDB: 4N4W; *Wang et al., 2014*).

| Modeled residues in 5L7D-inac-Apo-SMO | Constraints | Location |
|---|---|---|
| I429 | None | ICL3 |
| K430 | None | ICL3 |
| S431 | None | ICL3 |
| N432 | None | ICL3 |
| H433 | None | ICL3 |
| P434 | None | ICL3 |
| G435 | None | ICL3 |
| L436 | None | ICL3 |
| L437 | None | ICL3 |
| S438 | None | ICL3 |
| E439 | None | ICL3 |
| K440 | $\alpha$-helical | TM6 |
| A441 | $\alpha$-helical | TM6 |
| A442 | $\alpha$-helical | TM6 |
| S443 | $\alpha$-helical | TM6 |
| K444 | $\alpha$-helical | TM6 |
| I445 | $\alpha$-helical | TM6 |

## Lipid composition

**Appendix 1—table 2.** Composition of the membrane used for embedding the protein in simulations.

| Lipid Name | Upper Leaflet | Lower Leaflet |
|---|---|---|
| Cholesterol (CHL1) | 21 | 21 |
| 1-palmitoyl-2-oleoylphosphatidylcholine (POPC) | 76 | 76 |
| Palmitoylsphingomyelin (PSM) | 4 | 4 |
| Total | 101 | 101 |

## Distances used for trajectory featurization in adaptive sampling

**Appendix 1—table 3.** Adaptive Sampling metrics used for clustering during the iterative landscape exploration process.

| W281 | M525 | V321 | M525 | W281 | V321 |
|---|---|---|---|---|---|
| S278 | F526 | L325 | M525 | T241 | F274 |
| D473 | E518 | D384 | S387 | D384 | Y394 |
| Y322 | F391 | G277 | V321 | W281 | V321 |
| V319 | Y323 | V276 | I320 | I316 | V319 |
| N219 | P513 | L221 | I509 | Y487 | I509 |

*Appendix 1—table 3 Continued on next page*

*Appendix 1—table 3 Continued*

| W281 | M525 | V321 | M525 | W281 | V321 |
|------|------|------|------|------|------|
| S483 | N511 | Y207 | K395 | I215 | M301 |
| N219 | W480 | W109 | Y130 | W109 | R161 |
| E160 | R485 | I156 | E160 | L112 | I156 |
| N114 | V210 | D209 | A492 | E208 | V488 |
| CHL | H470 | CHL | N521 | CHL | F391 |
| CHL | Y394 | CHL | S387 | CHL | W281 |
| CHL | L522 | CHL | N521 | CHL | L325 |
| CHL | V404 | CHL | T466 | CHL | I408 |
| CHL | T528 | CHL | V329 | CHL | F462 |
| CHL | V463 | CHL | F274 | CHL | F332 |
| CHL | E518 | CHL | D473 | CHL | N521 |
| CHL | F391 | CHL | Y394 | CHL | R400 |
| CHL | N521 | CHL | L522 | CHL | W281 |
| CHL | M525 | CHL | P220 | CHL | F484 |
| CHL | S387 | CHL | H470 | CHL | D384 |
| CHL | L515 | CHL | V386 | CHL | K395 |
| CHL | N219 | CHL | F222 | CHL | W480 |
| CHL | P513 | CHL | L221 | CHL | W109 |
| CHL | R161 | CHL | I156 | CHL | D95 |
| CHL | Y130 | CHL | I496 | CHL | L112 |
| CHL | N114 | CHL | K105 | CHL | F166 |
| CHL | V210 | CHL | L489 | CHL | V157 |
| CHL | E160 | CHL | L108 | CHL | P164 |

## Round-wise data collection cholesterol translocation

**Appendix 1—table 4.** Round-wise data collection for Cholesterol transport of SMO.

| Simulation Round | Amount of Data |
|------------------|----------------|
| Round 1 | 20.38 $\mu$S |
| Round 2 | 460.15 $\mu$S |
| Round 3 | 71.12 $\mu$S |
| Round 4 | 82.17 $\mu$S |
| Round 5 | 43.26 $\mu$S |
| Round 6 | 26.70 $\mu$S |
| Round 7 | 508.16 $\mu$S |
| Round 8 | 184.63 $\mu$S |
| Round 9 | 118.69 $\mu$S |
| Round 10 | 550.59 $\mu$S |
| Total | 2066.227 $\mu$S |

## Appendix 2

## Roundwise data collection for SMO

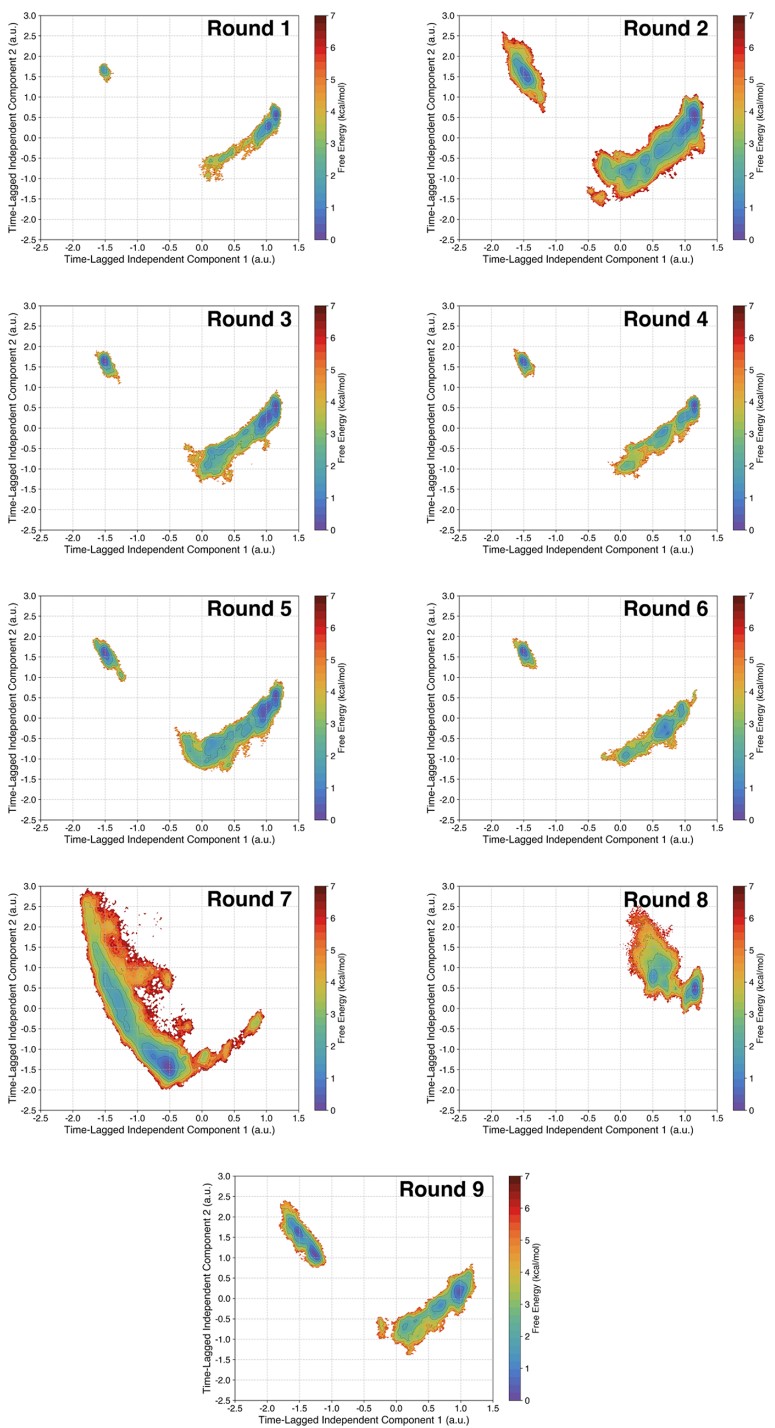

**Appendix 2—figure 1.** Roundwise data collection for SMO Cholesterol transport as projected along the first two time-lagged independent components (tICs). Round numbers are specified in the respective plots.

**MSM Construction**

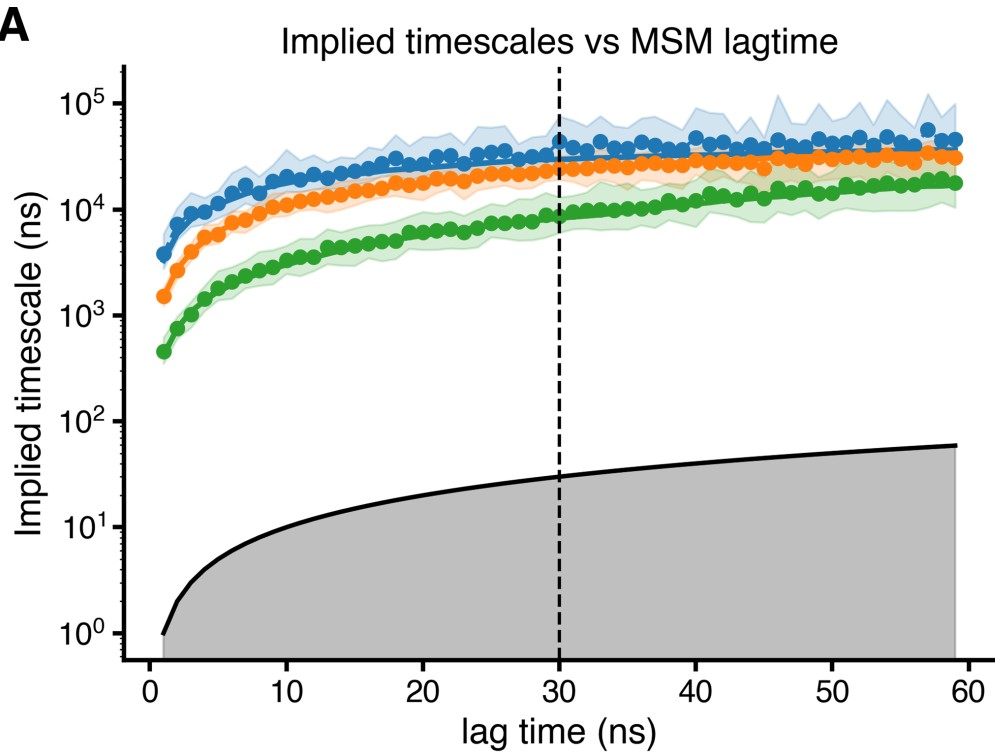

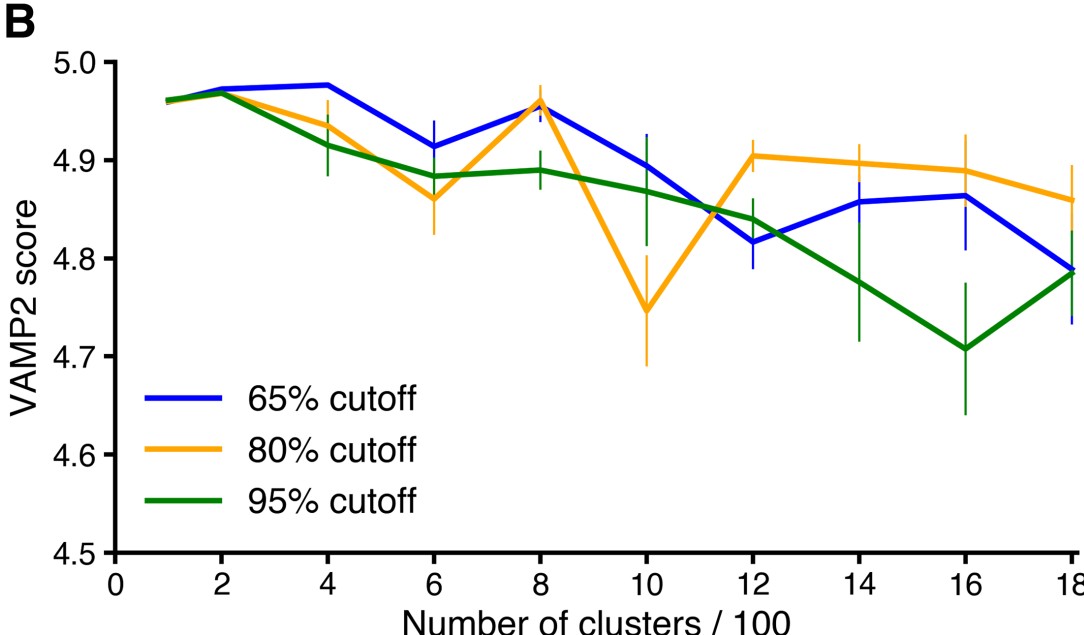

**Appendix 2—figure 2.** MSM Construction for SMO Cholesterol transport - Pathway 1. (**A**) Implied Timescales versus MSM Lagtime plot shows the convergence of timescales. A lag time of 30 ns was chosen for the MSM construction. (**B**) VAMP2 score versus Number of Clusters used for clustering the TICA-reduced data at three different variational cutoffs. The final MSM was made using 200 clusters and a 95% cutoff (corresponding to 42 tIC dimensions).

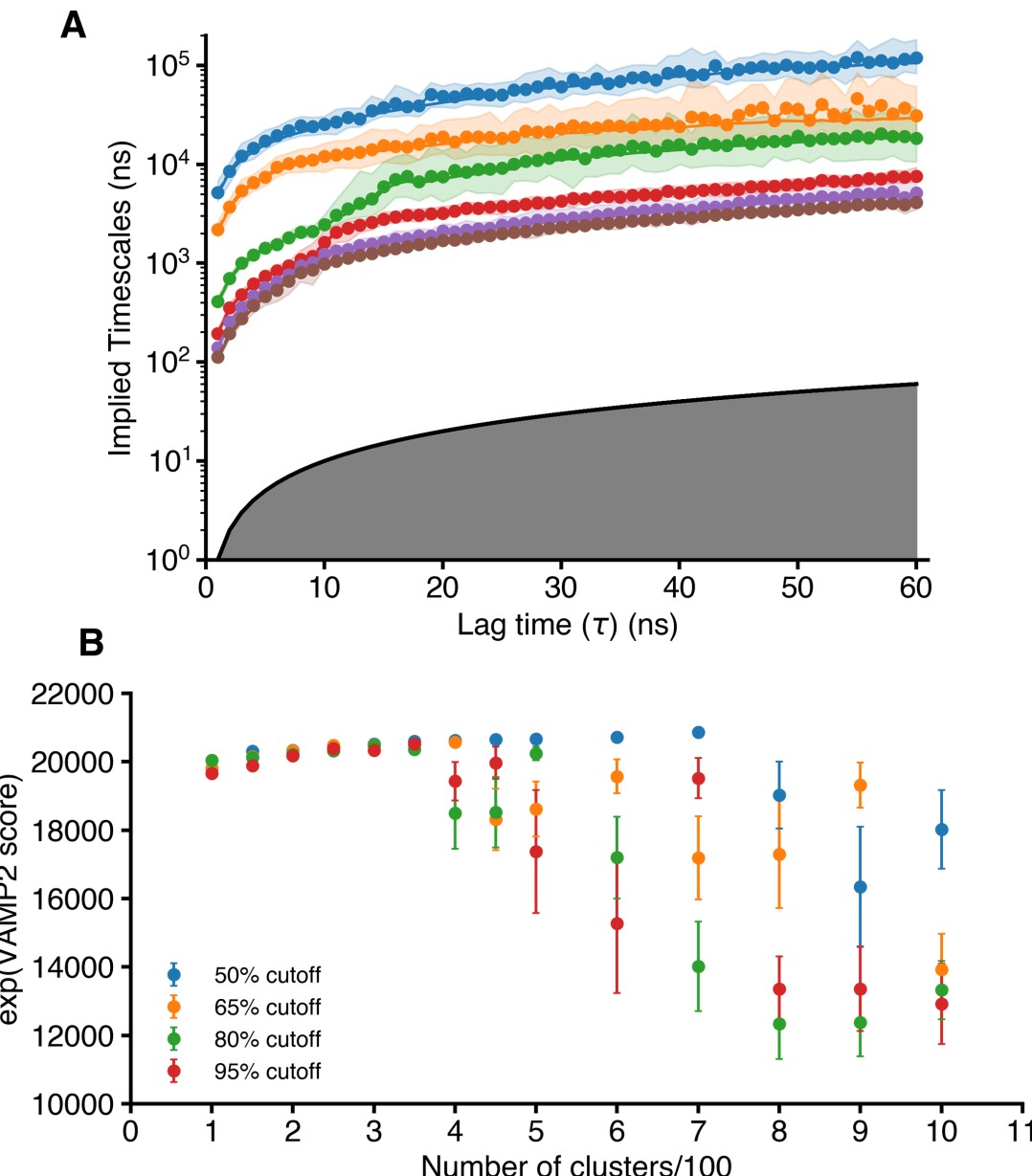

**Appendix 2—figure 3.** MSM Construction for SMO Cholesterol transport - Pathway 2. (**A**) The Implied Timescales versus MSM Lagtime plot shows the convergence of timescales. A lag time of 30 ns was chosen for the MSM construction. (**B**) VAMP2 score versus Number of Clusters used for clustering the TICA-reduced data at three different variational cutoffs. The final MSM was made using 400 clusters and a 65% cutoff (corresponding to 28 tIC dimensions).

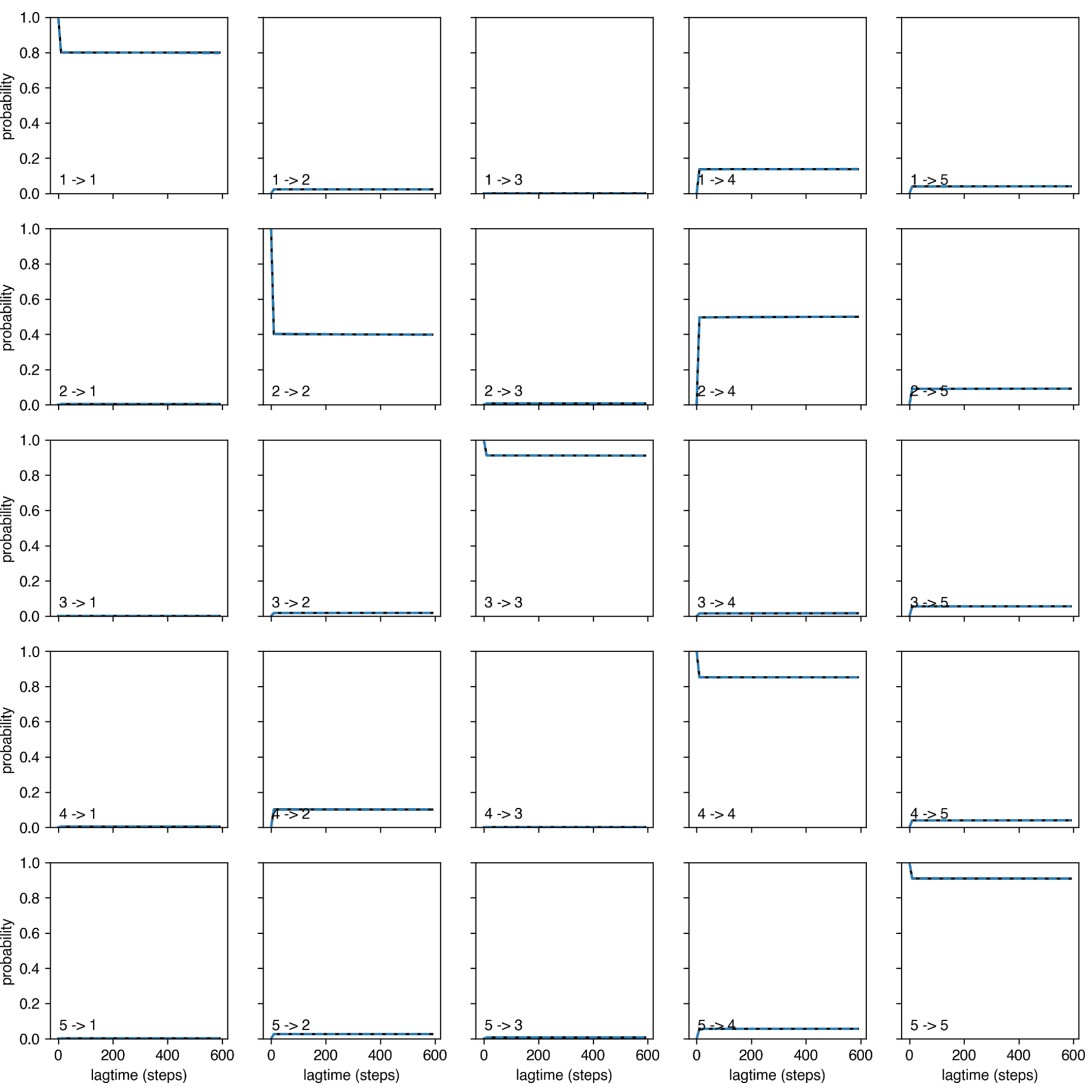

**Appendix 2—figure 4.** Chapman Kolmogorov Test for MSM validation - Pathway 1. Chapman-Kolmogorov test was performed using the deeptime library (*Hoffmann et al., 2022*).

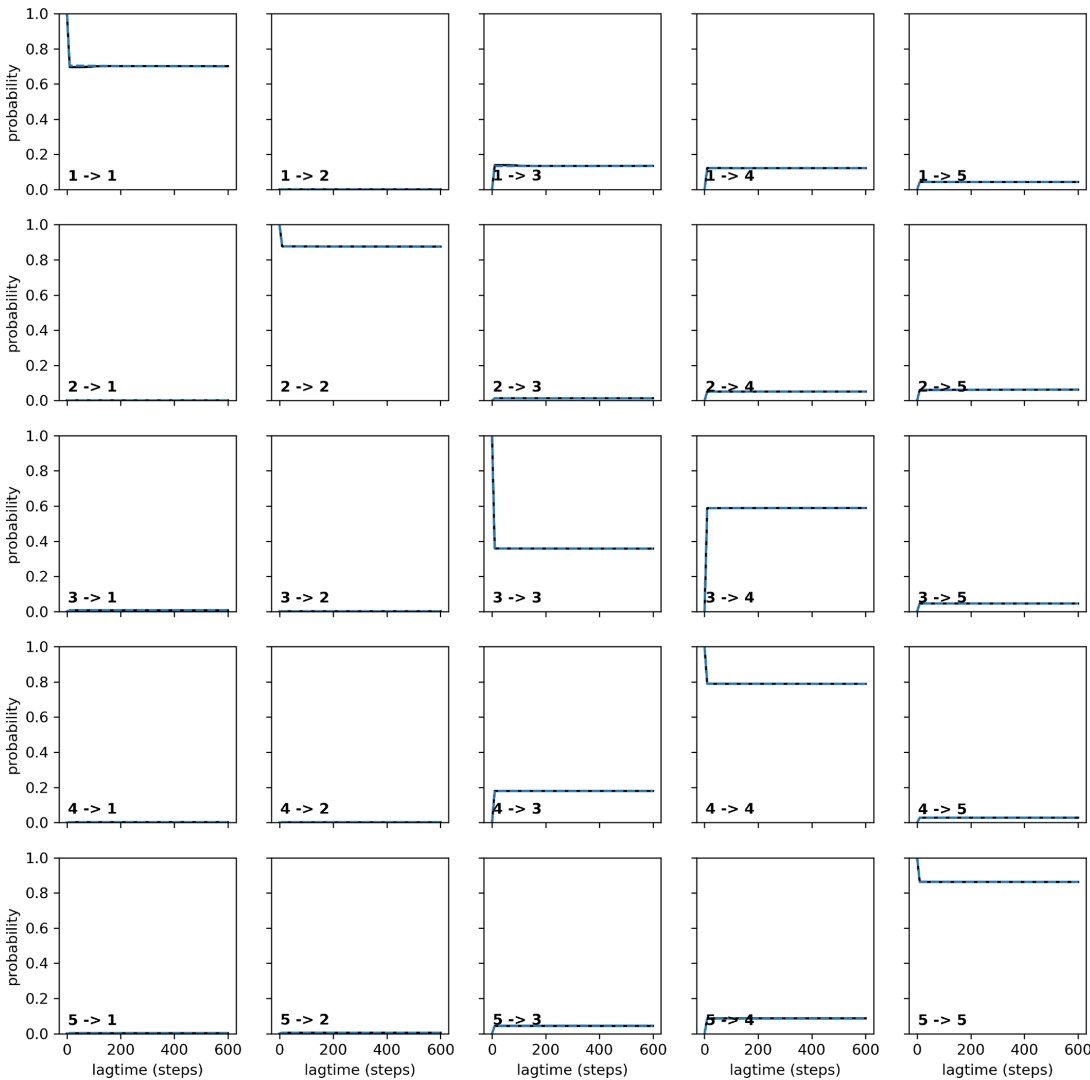

**Appendix 2—figure 5.** Chapman-Kolmogorov Test for MSM validation - Pathway 2. The Chapman-Kolmogorov test was performed using the deeptime library (*Hoffmann et al., 2022*).

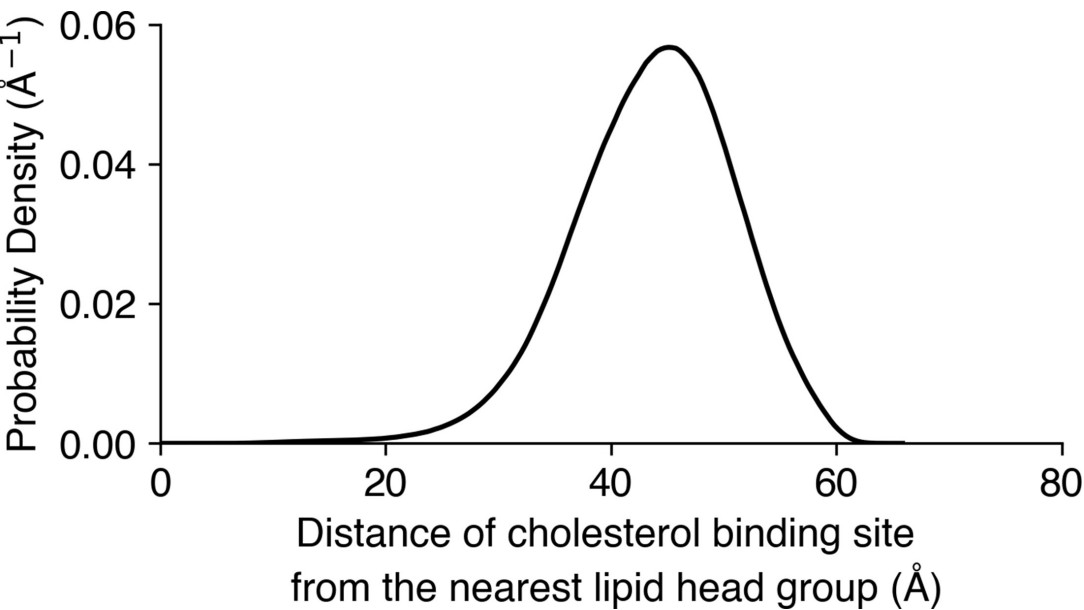

**Appendix 2—figure 6.** Distance of the binding site of cholesterol (marked by D95) from the nearest lipid headgroups in the membrane. Kernel density plotted for 2.1 ms of simulation data. The binding site is at least 20 Å away from the nearest lipid headgroup in simulations.

