## [Editor Report · eLife Assessment]

In this **important** study, the authors conducted extensive sets of computational and investigations of the mechanism of cholesterol transport in the smoothened (SMO) protein. The computational component integrated multiple state-of-the-art approaches such as adaptive sampling, free energy simulations, and Markov state modeling, providing **compelling** support for the proposed mechanistic model, which is further validated with **solid** experimental mutagenesis data.

---

## [Referee Report · Reviewer #1 (Public review)]

Summary:

This manuscript uses primarily simulation tools to probe the pathway of cholesterol transport with the smoothened (SMO) protein. The pathway to the protein and within SMO is clearly discovered and interactions deemed important are tested experimentally to validate the model predictions.

Strengths:

The authors have clearly demonstrated how cholesterol might go from the membrane through SMO for the inner and outer leaflets of a symmetrical membrane model. The free energy profiles, structural conformations and cholesterol-residue interactions are clearly described.

Weaknesses:

None. I find the revised manuscript strong and the work should be published.

---

## [Referee Report · Reviewer #2 (Public review)]

Summary:

In this work, the authors applied a range of computational methods to probe the translocation of cholesterol through the Smoothened receptor. They test whether cholesterol is more likely to enter the receptor straight from the outer leaflet of membrane or via a binding pathway in the inner leaflet first. Their data reveal that both pathways are plausible but that the free energy barriers of pathway 1 is lower suggesting this route is preferable. They also probe the pathway of cholesterol transport from the transmembrane region to the cysteine-rich domain (CRD).

Strengths:

A wide range of computational techniques are used, including potential of mean force calculations, adaptative sampling, dimensionality reduction using tICA, and MSM modelling. These are all applied in a rigorous manner and the data are very convincing. The computational work is an exemplar of a well-carried out study.

Their computational predictions are experimentally supported using mutagenesis, with an excellent agreement between their PMF and mRNA fold change data.

The data are described clearly and coherently, with excellent use of figures. They combine their findings into a mechanism for cholesterol transport, which on the whole seems sound.

Their methods are described well, and much of their analysis methods have been made available via GitHub, which is an additional strength.

---

## [Referee Report · Reviewer #3 (Public review)]

This manuscript presents a study combining molecular dynamics simulations and Hedgehog (Hh) pathway assays to investigate cholesterol translocation pathways to Smoothened (SMO), a G protein-coupled receptor central to Hedgehog signal transduction. The authors identify and characterize two putative cholesterol access routes to the transmembrane domain (TMD) of SMO and propose a model whereby cholesterol traverses through the TMD to the cysteine-rich domain (CRD), which is presented as the primary site of SMO activation.

The MD simulations and biochemical experiments are carefully executed and provide useful data.

Comments on revisions:

I appreciate the authors' detailed response and the substantial revisions made to the manuscript. The changes addressing Comments 3.1-3.5 have significantly improved the balance and framing of the work, and my primary concerns regarding overstatement and selective interpretation have been satisfactorily addressed.

The authors' rebuttal to my initial review includes extended argumentation regarding specific interpretations of prior studies and broader models of SMO regulation. These issues represent longstanding differences in interpretation that have already been discussed extensively in the literature and are not essential to evaluating the quality or conclusions of the present study.

For readers seeking a comprehensive and balanced overview of cholesterol-dependent SMO activation that integrates both CRD- and TMD-centered models, I would point to recent review articles (e.g., Zhang and Beachy, Nat Rev Mol Cell Biol2023). I do not feel it is productive to rehash these debates further in the context of this review, and I have no additional substantive concerns with the revised manuscript.

---

## [Author Response]

The following is the authors’ response to the original reviews

**Public Reviews:**

**Reviewer #1 (Public review):**
Summary:This manuscript uses primarily simulation tools to probe the pathway of cholesterol transport with the smoothened (SMO) protein. The pathway to the protein and within SMO is clearly discovered, and interactions deemed important are tested experimentally to validate the model predictions.Strengths:The authors have clearly demonstrated how cholesterol might go from the membrane through SMO for the inner and outer leaflets of a symmetrical membrane model. The free energy profiles, structural conformations, and cholesterol-residue interactions are clearly described.

We thank the reviewer for their kind words.

(1) Membrane Model: The authors decided to use a rather simple symmetric membrane with just cholesterol, POPC, and PSM at the same concentration for the inner and outer leaflets. This is not representative of asymmetry known to exist in plasma membranes (SM only in the outer leaflet and more cholesterol in this leaflet). This may also be important to the free energy pathway into SMO. Moreover, PE and anionic lipids are present in the inner leaflet and are ignored. While I am not requesting new simulations, I would suggest that the authors should clearly state that their model does not consider lipid concentration leaflet asymmetry, which might play an important role.

We thank the reviewer for their comment. Membrane asymmetry is inherent in endogenous systems; we acknowledge that as a limitation of our current model. We have addressed the comment by adding this limitation to our discussion in the manuscript.

Added lines: (End of paragraph 6, Results subsection 2):

“One possibility that might alter the thermodynamic barriers is native membrane asymmetry, particularly the anionic lipid-rich inner leaflet. This presents as a limitation of our current model.”

(2) Statistical comparison of barriers: The barriers for pathways 1 and 2 are compared in the text, suggesting that pathway 2 has a slightly higher barrier than pathway 1. However, are these statistically different? If so, the authors should state the p-value. If not, then the text in the manuscript should not state that one pathway is preferred over the other.

We thank the reviewer for their comment. We have added statistical t-tests for the barriers.

Changes made: (Paragraph 6, Results subsection 2)

“However, we also observe that pathway 1 shows a lower thermodynamic barrier (5.8 *±* 0.7 kcal/mol v/s 6.5 *±* 0.8 kcal/mol, p = 0.0013)”

(3) Barrier of cholesterol (reasoning): The authors on page 7 argue that there is an enthalpy barrier between the membrane and SMO due to the change in environment. However, cholesterol lies in the membrane with its hydroxyl interacting with the hydrophilic part of the membrane and the other parts in the hydrophobic part. How is the SMO surface any different? It has both characteristics and is likely balanced similarly to uptake cholesterol. Unless this can be better quantified, I would suggest that this logic be removed.

We thank the reviewer for this suggestion. We have removed the line to avoid confusion.

**Reviewer #2 (Public review):**
Summary:In this work, the authors applied a range of computational methods to probe the translocation of cholesterol through the Smoothened receptor. They test whether cholesterol is more likely to enter the receptor straight from the outer leaflet of the membrane or via a binding pathway in the inner leaflet first. Their data reveal that both pathways are plausible but that the free energy barriers of pathway 1 are lower, suggesting this route is preferable. They also probe the pathway of cholesterol transport from the transmembrane region to the cysteine-rich domain (CRD).Strengths:(1) A wide range of computational techniques is used, including potential of mean force calculations, adaptive sampling, dimensionality reduction using tICA, and MSM modelling. These are all applied rigorously, and the data are very convincing. The computational work is an exemplar of a well-carried out study.(2) The computational predictions are experimentally supported using mutagenesis, with an excellent agreement between their PMF and mRNA fold change data.(3) The data are described clearly and coherently, with excellent use of figures. They combine their findings into a mechanism for cholesterol transport, which on the whole seems sound.(4) The methods are described well, and many of their analysis methods have been made available via GitHub, which is an additional strength.Weaknesses:(1) Some of the data could be presented a little more clearly. In particular, Figure 7 needs additional annotation to be interpretable. Can the position of the cholesterol be shown on the graph so that we can see the diameter change more clearly?

We thank the reviewer for this suggestion. We have added the cholesterol positions as requested.

Changes made: (Caption, Figure 7)

“The tunnel profile during cholesterol translocation in SMO. (a) Free energy plot of the zcoordinate v/s the tunnel diameter when cholesterol is present in the core TMD. The tunnel shows a spike in the radius in the TMD domain, indicating the presence of a cholesterol-accommodating cavity. (b) Representative figure for the tunnel when a cholesterol molecule is in the TMD. (c) Same as (a), when cholesterol is at the TMD-CRD interface. (e) same as (b), when cholesterol is at the TMD-CRD interface. (e) same as (a), when cholesterol is at the CRD binding site. (f) same as (b), when cholesterol is at the CRD binding site. Tunnel diameters shown as spheres. Cholesterol positions marked on plots using dotted lines. All snapshots presented are frames taken from MD simulations.”

(2) In Figure 3C, it doesn’t look like the Met is constricting the tunnel at all. What residue is constricting the tunnel here? Can we see the Ala and Met panels from the same angle to compare the landscapes? Or does the mutation significantly change the tunnel? Why not A283 to a bulkier residue? Finally, the legend says that the figure shows that cholesterol can still pass this residue, but it doesn’t really show this. Perhaps if the HOLE graph was plotted, we could see the narrowest point of the tunnel and compare it to the size of cholesterol.

We thank the reviewer for this suggestion. A283 was mutated to methionine as it presents with a longer heavy tail containing sulfur. We have plotted the tunnel radii for both WT and A283M mutants and added them as a supplemental figure. As shown in the figure, the presence of methionine doesn’t completely block the tunnel, but occludes it, thereby increasing the barrier for cholesterol transport slightly.

Changes made: (End of Results subsection 1)

“When we calculated the PMF for cholesterol entry, A^2.60*f*^M mutant showed restricted tunnel but it did not fully block the tunnel (Figure 3—figure Supplement 3).”

(3) The PMF axis in 3b and d confused me for a bit. Looking at the Supplementary data, it’s clear that, e.g., the F455I change increases the energy barrier for chol entering the receptor. But in 3d this is shown as a -ve change, i.e., favourable. This seems the wrong way around for me. Either switch the sign or make this clearer in the legend, please.

We thank the reviewer for this suggestion. We measured ∆*PMF* as *PMF_WT_ PMF_mutant_*, hence the negative values. We have added additional text to the legend to clarify this.

Changes made: (Caption, Figure 3)

“(b) ∆*Gli1* mRNA fold change (high SHH vs untreated) and ∆ PMF (difference of peak PMF , calculated as *PMF_WT_* - *PMF_mutant_*) plotted for the mutants in Pathway 1. (c) Example mutant A^2_._60*f*^M shows that cholesterol can enter SMO through Pathway 1 even on a bulky mutation. (d) Same as (b) but for Pathway 2 (e) Example mutant L^5.62*f*^A shows that cholesterol can enter SMO through Pathway 2 due to lesser steric hindrance. All snapshots presented are frames taken from MD simulations.”

Changes made: (Caption, Figure 6)

“(b) ∆*Gli1* mRNA fold change (high SHH vs untreated) and ∆ PMF (difference of peak PMF, calculated as *PMF_WT_* - *PMF_mutant_*) plotted for mutants along the TMD-CRD pathway. (c, d) Example mutants Y*LD*A and F^5.65*f*^A show that cholesterol is unable to translocate through this pathway because of the loss of crucial hydrophobic contacts provided by Y207 and F484 and along the solvent-exposed pathway.”

(4) The impact of G280V is put down to a decrease in flexibility, but it could also be a steric hindrance. This should be discussed.

We thank the reviewer for this suggestion. We have added it as a possible mechanism of the decrease in activity of SMO.

Changes made: (Paragraph 5, Results subsection 1)

“We mutated G280^2.57*f*^ to valine - G^2.57*f*^V to test whether reducing the flexibility of TM2 prevents cholesterol entry into the TMD. Consequently, the activity of *mSMO* showed a decrease. However, this decrease could also be attributed to steric hindrance added by the presence of a bulky propyl group in valine.”

(5) Are the reported energy barriers of the two pathways (5.8plus minus0.7 and 6.5plus minus0.8 kcal/mol) significantly and/or substantially different enough to favour one over the other? This could be discussed in the manuscript.

We thank the reviewer for this suggestion. We have added statistical t-tests for the barriers.

Changes made: (Paragraph 6, Results subsection 2)

“However, we also observe that pathway 1 shows a lower thermodynamic barrier (5.8 *±* 0.7 kcal/mol v/s 6.5 *±* 0.8 kcal/mol, p = 0.001)”

(6) Are the energy barriers consistent with a passive diffusion-driven process? It feels like, without a source of free energy input (e.g., ion or ATP), these barriers would be difficult to overcome. This could be discussed.

We thank the reviewer for this suggestion. We have added a discussion to further clarify this point.

Discussion: (Paragraph 6, Results subsection 2)

“These values are comparable to ATP-Binding Cassette (ABC) transporters of membrane lipids, which use ATP hydrolysis (-7.54 *±* 0.3 kcal/mol) (Meurer et al., 2017) to drive lipid transport from the membrane to an extracellular acceptor. Some of these transporters share the same mechanism as SMO, where the lipid from the inner leaflet is flipped and transported to the extracellular acceptor protein (Tarling et al., 2013). Additionally, for secondary active transporters that do not use ATP for the transport of substrates, a thermodynamic barrier of 5-6 kcal/mol has been reported in literature. (Chan et al., 2022; Selvam et al., 2019; McComas et al., 2023; Thangapandian et al., 2025).”

(7) Regarding the kinetics from MSM, it is stated that the values seen here are similar to MFS transporters, but this then references another MSM study. A comparison to experimental values would support this section a lot.

We thank the reviewer for this suggestion. We have added a discussion discussing millisecond-scale timescales measured for MFS transporters.

Changes made: (Paragraph 2, Results subsection 5)

“These timescales are comparable to the substrate transport timescales of Major Facilitator Superfamily (MFS) transporters (Chan et al., 2022). Furthermore, several experimental studies have also resolved the millisecond-scale kinetics of MFS transporters (Blodgett and Carruthers, 2005; Körner et al., 2024; Bazzone et al., 2022; Smirnova et al., 2014; Zhu et al., 2019), further corroborating the results from our study.”

**Reviewer #2 (Recommendations for the authors):**
(1) The heatmaps in Figures 2a and 4a are great. On these, an arrow denotes what looks like a minimum energy path. Is it possible to see this plotted, as this might show the height of the energy barriers more clearly?

We thank the reviewer for this suggestion. We have computed the minimum energy paths for both pathways and presented them in a supplementary figure.

Added lines: (Paragraph 4, Results subsection 1):

For further clarity, we have plotted the minimum energy path taken by cholesterol as it translocates along this pathway (Figure 2—figure Supplement 3a,b)

Added lines: (Paragraph 4, Results subsection 2):

For further clarity, we have plotted the minimum energy path taken by cholesterol as it translocates along this pathway (Figure 2—figure Supplement 3c,d)

(2) The tiCA data in S15 is first referred to on line 137, but the technique isn’t introduced until line 222. This makes understanding the data a little confusing. Reordering this might improve readability.

We thank the reviewer for this suggestion. We have reordered the text to make it clearer.

Changes made: (Paragraph 2, Results subsection 1) This provides evidence for multiple stable poses along the pathway as observed in the multiple stable poses of cholesterol in Cryo-EM structures of SMO bound to sterols (Deshpande et al., 2019; Qi et al., 2019b, 2020). A reliable estimate of the barriers comes from using the time-lagged Independent Components (tICs), which project the entire dataset along the slowest kinetic degrees of freedom. Overall, the highest barrier along Pathway 1 is *∼* 5.8 *±* 0.7 kcal/mol, and it is associated with the entry of cholesterol into the TMD (Figure 2—Figure Supplement 2).

Changes made: (Paragraph 3, Results subsection 2)

“On plotting the first two components of tICs, (Figure 2—Figure Supplement 2), we observe that the energetic barrier between *η* and *θ* is ∼6.5 *±* 0.8 kcal/mol.”

(3) Missing bracket on line 577.

We thank the reviewer for this suggestion. The typo has been fixed.

(4) Line 577: Fig. S2nd?

We thank the reviewer for this suggestion. This typo has been fixed.

**Reviewer #3 (Public review):**
Summary:This manuscript presents a study combining molecular dynamics simulations and Hedgehog (Hh) pathway assays to investigate cholesterol translocation pathways to Smoothened (SMO), a G protein-coupled receptor central to Hedgehog signal transduction. The authors identify and characterize two putative cholesterol access routes to the transmembrane domain (TMD) of SMO and propose a model whereby cholesterol traverses through the TMD to the cysteine-rich domain (CRD), which is presented as the primary site of SMO activation. The MD simulations and biochemical experiments are carefully executed and provide useful data.Weaknesses:However, the manuscript is significantly weakened by a narrow and selective interpretation of the literature, overstatement of certain conclusions, and a lack of appropriate engagement with alternative models that are well-supported by published data-including data from prior work by several of the coauthors of this manuscript. In its current form, the manuscript gives a biased impression of the field and overemphasizes the role of the CRD in cholesterol-mediated SMO activation. Below, I provide specific points where revisions are needed to ensure a more accurate and comprehensive treatment of the biology.(1) Overstatement of the CRD as the Orthosteric Site of SMO ActivationThe manuscript repeatedly implies or states that the CRD is the orthosteric site of SMO activation, without adequate acknowledgment of alternative models. To give just a few examples (of many in this manuscript):(a) “PTCH is proposed to modulate the Hh signal by decreasing the ability of membrane cholesterol to access SMO’s extracellular cysteine-rich domain (CRD)” (p. 3).(b) “In recent years, there has been a vigorous debate on the orthosteric site of SMO” (p. 3).(c) “cholesterol must travel through the SMO TMD to reach the orthosteric site in the CRD” (p. 4).(d) “we observe cholesterol moving along TM6 to the TMD-CRD interface (common pathway, Fig. 1d) to access the orthosteric binding site in the CRD” (p. 6).While the second quote in this list at least acknowledges a debate, the surrounding text suggests that this debate has been entirely resolved in favor of the CRD model. This is misleading and not reflective of the views of other investigators in the field (see, for example, a recent comprehensive review from Zhang and Beachy, Nature Reviews Molecular and Cell Biology 2023, which makes the point that both the CRD and 7TM sites are critical for cholesterol activation of SMO as well as PTCH-mediated regulation of SMO-cholesterol interactions).In contrast, a large body of literature supports a dual-site model in which both the CRD and the TMD are bona fide cholesterol-binding sites essential for SMO activation. Examples include:(a) Byrne et al., Nature 2016: point mutation of the CRD cholesterol binding site impairs-but does not abolish-SMO activation by cholesterol (SMO D99A, Y134F, and combination mutants - Fig 3 of the 2016 study).(b) Myers et al., Dev Cell 2013 and PNAS 2017: CRD deletion mutants retain responsiveness to PTCH regulation and cholesterol mimetics (similar Hh responsiveness of a CRD deletion mutant is also observed in Fig. 4 Byrne et al, Nature 2016).(c) Deshpande et al., Nature 2019: mutation of residues in the TMD cholesterol binding site blocks SMO activation entirely, strongly implicating the TMD as a required site, in contrast to the partial effects of mutating or deleting the CRD site.Qi et al., Nature 2019, and Deshpande et al., Nature 2019, both reported cholesterol binding at the TMD site based on high-resolution structural data. Oddly, Deshpande et al., Nature 2019, is not cited in the discussion of TMD binding on p. 3, despite being one of the first papers to describe cholesterol in the TMD site and its necessity for activation (the authors only cite it regarding activation of SMO by synthetic small molecules).Kinnebrew et al., Sci Adv 2022 report that CRD deletion abolished PTCH regulation, which is seemingly at odds with several studies above (e.g., Byrne et al, Nature 2016; Myers et al, Dev Cell 2013); but this difference may reflect the use of an N-terminal GFP fusion to SMO in the Kinnebrew et al 2022, which could alter SMO activation properties by sterically hindering activation at the TMD site by cholesterol (but not synthetic SMO agonists like SAG); in contrast, the earlier work by Byrne et al is not subject to this caveat because it used an untagged, unmodified form of SMO.Although overexpression of PTCH1 and SMO (wild-type or mutant) has been noted as a caveat in studies of CRD-independent SMO activation by cholesterol, this reviewer points out that several of the studies listed above include experiments with endogenous PTCH1 and low-level SMO expression, demonstrating that SMO can clearly undergo activation by cholesterol (as well as regulation by PTCH1) in a manner that does not require the CRD.Recommendation: The authors should revise the manuscript to provide a more balanced overview of the field and explicitly acknowledge that the CRD is not the sole activation site. Instead, a dual-site model is more consistent with available structural, mutational, and functional data. In addition, the authors should reframe their interpretation of their MD studies to reflect this broader and more accurate view of how cholesterol binds and activates SMO.

We thank the reviewer for this comprehensive overview of the existing literature. We agree that cholesterol binding to both the TMD and CRD sites is required for full activation of SMO. As described below in responses to comments, we have made changes to the manuscript to make this point clear. For instance, in the revised manuscript, we refrain from calling the CRD cholesterol binding site the “orthosteric site”. Instead, we highlight that the goal of the manuscript is not to resolve the debate over whether the TMD or CRD site is more important for PTCH1 regulation by SMO but rather to use molecular dynamics to understand the fascinating question of how cholesterol in the membrane can reach the CRD, located at a significant distance above the outer leaflet of the membrane. We believe that this is an important goal since there is an abundance of evidence that supports the view that PTCH1 inhibits SMO by reducing cholesterol access to the CRD. This evidence is now summarized succinctly in the introduction:

Changes made: (Paragraph 4, Introduction)

“While cholesterol binding to both the TMD and CRD sites is required for full SMO activation, our work focuses on how cholesterol gains access to the CRD site, perched above the outer leaflet of the membrane (Luchetti et al., 2016; Kinnebrew et al., 2022). Multiple lines of evidence suggest that PTCH1-regulated cholesterol binding to the CRD plays an instructive role in SMO regulation both in cells and animals. Mutations in residues predicted to make hydrogen bonds with the hydroxyl group of cholesterol bound to the CRD reduced both the potency and efficacy of SHH in cellular signaling assays (Kinnebrew et al., 2022; Byrne et al., 2016) and, more importantly, eliminated HH signaling in mouse embryos (Xiao et al., 2017). Experiments using both covalent and photocrosslinkable sterol probes in live cells directly show that PTCH1 activity reduces sterol access to the CRD (Kinnebrew et al., 2022; Xiao et al., 2017). Notably, our simulations evaluate a path of cholesterol translocation that includes both the TMD and CRD sites: cholesterol first enters the 7-transmembrane domain bundle from the membrane; it then engages the TMD site before continuing along a conduit to the CRD site. Thus, we analyze translocation energetics and residue-level contacts along a path that includes both the TMD and the CRD.”

However, Reviewer 3 makes several comments below that are biased, inaccurate, or selective. We feel it is important to address these so readers can approach the literature from a balanced perspective. Indeed, the *eLife* review forum provides an ideal venue to present contrasting views on a scientific model. We encourage the editors to publish both Reviewer 3’s comments and our response in full so readers can read the original papers and reach their own conclusions. It is important to note these issues are not relevant to the quality of the computational and experimental data presented in this paper.

We have now removed the term “orthosteric” to describe the CRD site throughout the paper and clearly state in the introduction that “both the CRD and TMD sites are required for SMO activation” but that our focus is on how cholesterol moves from the membrane to the CRD site. There is no doubt that cholesterol binding to the CRD plays a key role in SMO activation– our focus on this path is justified and does not devalue the importance of the TMD site. Our prior models (see Figure 7 of Kinnebrew 2022 explicitly include contributions of both sites).

Now we respond to some of the concerns outlined, individually:

(1) Byrne et al., Nature 2016: point mutation of the CRD cholesterol binding site impairs-but does not abolish-SMO activation by cholesterol (SMO D99A, Y134F, and combination mutants - Fig 3 of the 2016 study)

The fact that a point mutation dramatically diminishes (but does not abolish signaling) does not mean that the CRD cholesterol binding site is not important for SMO regulation. Indeed, the reviewer fails to mention that Song et. al. (Molecular Cell, 2017) found that a SMO protein carrying a subtle mutation at D99 (D95/99N, a residue that makes a hydrogen bond with the cholesterol hydroxyl) completely abolishes SMO signaling in mouse embryos. Thus, the CRD site is critical for SMO activation in an intact animal, justifying our focus on evaluating the path of cholesterol translocation to the CRD site.

(2) Myers et al., Dev Cell 2013 and PNAS 2017: CRD deletion mutants retain responsiveness to PTCH regulation and cholesterol mimetics (similar Hh responsiveness of a CRD deletion mutant is also observed in Fig 4 Byrne et al, Nature 2016).

The Reviewer fails to note that CRD-deleted versions of SMO have markedly (>10-fold) higher basal (i.e. ligand-independent) activity compared to full-length SMO. The response to SHH is minimal (∼2-fold), compared to >50-100-fold with full-length SMO. Thus, CRD-deleted SMO is likely in a non-native conformation. Local changes in cholesterol accessibility caused by PTCH1 inactivation or cholesterol loading can cause small fluctuations in delta-CRD activity, but this cannot be used to infer meaningful insights about how native, full-length SMO (with >10-fold lower basal activity) is regulated. We encourage the reviewer to read our previous paper (Kinnebrew et. al. 2022), which presents a unified view of how the TMD and CRD sites together regulate SMO activation.

A more physiological experiment, reported in Kinnebrew et. al. 2022, tested mutations in residues that make hydrogen bonds with cholesterol at the CRD and TMD sites in the context of full-length SMO. These mutants were stably expressed at moderate levels in Smo*−/−* cells. Mutations at the CRD site reduced the fold-increase in signaling output in response to SHH, as would be expected for a PTCH1-regulated site. In contrast, analogous mutations in the TMD site reduced the magnitude of both basal and maximal signaling, without affecting the fold-change in response to SHH. In signaling assays, the key parameter in evaluating the impact of a mutation is whether it impacts the change in output in response to a signal (in this case PTCH1 inactivation by SHH). A mutation in SMO that affects PTCH1 regulation is expected to decrease the fold-change in signaling in response to SHH, a criterion that is fulfilled by mutations in the CRD site. Accordingly, mutations in the CRD site abolish SMO signaling in mouse embryos (Xiao et al., 2017).

(3) Deshpande et al., Nature 2019: mutation of residues in the TMD cholesterol binding site blocks SMO activation entirely, strongly implicating the TMD as a required site, in contrast to the partial effects of mutating or deleting the CRD site.

Introduction of bulky mutations at the TMD site (V333F) that abolish SMO activity were first reported by Byrne et. al. 2016 and were used to markedly increase the stability of SMO for protein expression. These mutations indeed stabilize the inactive state of SMO, increasing protein abundance and completely preventing its localization at primary cilia. SMO variants carrying such bulky mutations cannot be used to infer the importance of the TMD site since they do not distinguish between the following possibilities: (1) SMO is inactive because the sterol cannot bind, or (2) SMO is inactive because it is locked in an inactive conformation, or (3) SMO is inactive because it cannot localize to primary cilia (where it must be localized to activate downstream signaling).

As described in Response 3.3, a better evaluation of the importance of the TMD site is the use of mutations in residues that make hydrogen bonds with the hydroxyl group of TMD cholesterol. These mutations do not markedly increase protein stability or prevent ciliary localization (Kinnebrew 2022, Fig.S2). While a TMD site mutation decreases the magnitude of maximal (and basal) SMO signaling, it does not impact the fold-increase in signal output in response to Hh ligands (the key parameter that should be used to evaluate PTCH1 activity).

(4) Qi et al., Nature 2019, and Deshpande et al., Nature 2019, both reported cholesterol binding at the TMD site based on high-resolution structural data. Oddly, Deshpande et al., Nature 2019 not cited in the discussion of TMD binding on p. 3, despite being one of the first papers to describe cholesterol in the TMD site and its necessity for activation (the authors only cite it regarding activation of SMO by synthetic small molecules)

The reference has now been added at this location in the manuscript.

(5) Kinnebrew et al., Sci Adv 2022 report that CRD deletion abolished PTCH regulation, which is seemingly at odds with several studies above (e.g., Byrne et al, Nature 2016; Myers et al, Dev Cell 2013); but this difference may reflect the use of an N-terminal GFP fusion to SMO in the Kinnebrew et al 2022, which could alter SMO activation properties by sterically hindering activation at the TMD site by cholesterol (but not synthetic SMO agonists like SAG); in contrast, the earlier work by Byrne et al is not subject to this caveat because it used an untagged, unmodified form of SMO.

The reviewer fails to note that CRD deleted versions of SMO have markedly (>10-fold) higher basal activity than full-length SMO. The response to SHH is minimal (∼2fold), compared to >50-fold with full-length SMO. Thus, CRD-deleted SMO is likely in a non-native conformation. Local changes in cholesterol accessibility caused by PTCH1 inactivation or cholesterol loading can cause small fluctuations in delta-CRD activity, but this cannot be used to infer meaningful insights about how native, full-length SMO (with >10-fold lower basal activity) is regulated. Please see Response 3.3 for further details.

Reviewer 3 presents an incomplete picture of the extensive experiments reported in Kinnebrew et. al. to establish the functionality of YFP-tagged delta-CRD SMO. Most importantly, a TMDselective sterol analog (KK174) can fully activate YFP-tagged delta-CRD, showing conclusively that the YFP fusion does not block sterol access to the TMD site. The fact that this protein is nearly unresponsive to SHH highlights the critical role of the CRD-bound cholesterol in SMO regulation by PTCH1. Indeed, the YFP-tagged, CRD-deleted SMO was made purposefully to test the requirement of the CRD in a construct that had normal basal activity. Again, this data justifies the value of investigating the path of cholesterol movement from the membrane via the TMD site to the CRD.

(6) Although overexpression of PTCH1 and SMO (wild-type or mutant) has been noted as a caveat in studies of CRD-independent SMO activation by cholesterol, this reviewer points out that several of the studies listed above include experiments with endogenous PTCH1 and low-level SMO expression, demonstrating that SMO can clearly undergo activation by cholesterol (as well as regulation by PTCH1) in a manner that does not require the CRD.

This comment is inaccurate. The data presented in Deshpande et. al. (and prior work in Myers et. al.) used transient transfection to overexpress SMO in Smo*−/−* cells. At the individual cell level transient transfection produces expression levels that are markedly higher (10-1000-fold) than stable expression (in addition to being more variable). Most scientists would agree that stable expression (as used in Kinnebrew 2022) at a moderate expression level is a better system to compare mutant phenotypes, assess basal and activated signaling, and provide an accurate measure of the fold-change in signal output in response to SHH. Notably, introduction of a mutation in the CRD cholesterol binding site at the endogenous mouse Smo locus (an even better experiment than stable expression) leads to complete loss of SMO activity (PMID 28344083). This result again justifies our investigation of the pathway of cholesterol movement from the membrane to the CRD site.

We have changed the initial discussion and reflect a more general outlook.

Changes made: (Paragraph 1, Introduction)

“PTCH modulates the availability of accessible cholesterol at the primary cilium and thereby regulates SMO, with models invoking effects on both the CRD and 7TM pockets.”

Changes made: (Results subsection 3, paragraph 1)

“According to the dual-site model, to reach the binding site in the CRD (*ζ*), cholesterol translocate along the TMD-CRD interface from the TM binding site (*α∗*) is required.”

Added lines: (Paragraph 5, Results subsection 3):

“The computational investigation showed here covers the dual-site model, where cholesterol reaches the CRD site via binding to the TM binding site first. In comparison to the CRD site, the TM site is more stable by ∼ 2 kcal/mol (Figure 2—Figure Supplement 3b, d).”

Added lines: (Paragraph 2, Conclusions):

“Here we have explored the role the CRD-site plays in SMO activation. In addition, through simulating the CRD site-dependent SMO activation hypothesis, we have also simulated the TMD site-dependent activation. We show that the overall stability of cholesterol is higher than the CRD site by ∼ 2 kcal/mol.”

(2) Bias in Presentation of Translocation PathwaysThe manuscript presents the model of cholesterol translocation through SMO to the CRD as the predominant (if not sole) mechanism of activation. Statements such as: "Cholesterol traverses SMO to ultimately reach the CRD binding site" (p. 6) suggest an exclusivity that is not supported by prior literature in the field. Indeed, the authors’ own MD data presented here demonstrate more stable cholesterol binding at the TMD than at the CRD (p 17), and binding of cholesterol to the TMD site is essential for SMO activation. As such, it is appropriate to acknowledge that cholesterol may activate SMO by translocating through the TM5/6 tunnel, then binding to the TMD site, as this is a likely route of SMO activation in addition to the CRD translocation route they highlight in their discussion.The authors describe two possible translocation pathways (Pathway 1: TM2/3 entry to TMD; Pathway 2: TM5/6 entry and direct CRD transfer), but do not sufficiently acknowledge that their own empirical data support Pathway 2 as more relevant. Indeed, because their experimental data suggest Pathway 2 is more strongly linked to SMO activation, this pathway should be weighted more heavily in the authors’ discussion. In addition, Pathway 2 is linked to cholesterol binding to both the TMD and CRD sites (the former because the TMD binding site is at the terminus of the hydrophobic tunnel, the latter via the translocation pathway described in the present manuscript), so it is appropriate that Pathway 2 figures more prominently than Pathway 1 in the authors’ discussion.The authors also claim that "there is no experimental structure with cholesterol in the inner leaflet region of SMO TMD" (p 16). However, a structural study of apo-SMO from the Manglik and Cheng labs (Zhang et al., Nat Comm, 2022) identified a cholesterol molecule docked at the TM5/6 interface and also proposed a "squeezing" mechanism by which cholesterol could enter the TM5/6 pocket from the membrane. The authors do not consider this SMO conformation in their models, nor do they discuss the possibility that conformational dynamics at the TM5/6 interface could facilitate cholesterol flipping and translocation into the hydrophobic conduit, despite both possibilities having precedent in the 2022 empirical cryoEM structural analysis.Recommendation: The authors should avoid oversimplifying the SMO cholesterol activation process, either by tempering these claims or broadening their discussion to better reflect the complexity and multiplicity of cholesterol access and activation routes for SMO. They should also consider the 2022 apo-SMO cryoEM structure in their analysis of the TM5/6 translocation pathway.

We thank the reviewer for this comprehensive overview of the existing literature and parts we have missed to include in the discussion. We agree with the reviewer, since our data shows that both pathways are probable. Through our manuscript, we have avoided using a competitive approach (that one pathway dominates over the other). Instead, we have evaluated both pathways independently and presented a comparative rather than competitive overview of both pathways from our observations. While we agree that experimental evidence suggests the inner leaflet pathway is possible, we cannot discount the observations made in previous studies that support the outer leaflet pathway, particularly Hedger et al. (2019), Bansal et al. (2023), and Kinnebrew et al. (2021). Therefore, considering the reviewer’s comments have made the following changes:

(1) Added lines: (Paragraph 3, Conclusions):

“We show that the barriers associated with the pathway starting from the outer leaflet are lower by ∼0.7 kcal, (p=0.0013). We also provide evidence that cholesterol can enter SMO via both leaflets, considering that multiple computational and experimental studies have found cholesterol entry sites and activation modulation via the outer leaflet, between TM2TM3. This is countered by evidence from multiple experimental and computational studies corroborating entry via the inner leaflet, between TM5-TM6, including this study. Overall, we posit that cholesterol translocation from either pathway is feasible.”

(2)nChanges made: (Paragraph 6, Results subsection 2)

“Based on our experimental and computational data, we conclude that cholesterol translocation can happen via either pathway. This is supported on the basis of the following observations: mutations along pathway 2 affect SMO activity more significantly, and the presence of a direct conduit that connects the inner leaflet to the TMD binding site. In addition, a resolved structure of SMO in the presence of cholesterol shows a cholesterol situated at the entry point from the membrane into the protein between TM5 and TM6, in the inner leaflet. However, we also observe that pathway 1 shows a lower thermodynamic barrier (5.8 *±* 0.7 kcal/mol vs. 6.5 *±* 0.8 kcal/mol, *p* = 0.0013). Additionally, PTCH1 controls cholesterol accessibility in the outer leaflet. This shows that there is a possibility for transport from both leaflets. One possibility that might alter the thermodynamic barriers is native membrane asymmetry, particularly the anionic lipid-rich inner leaflet. This presents as a limitation of our current model.”

(3)nChanges made: (Paragraph 1, Results subsection 2)

“In a structure resolved in 2022, cholesterol was observed at the interface between the protein and the membrane, in the inner leaflet, between TMs 5 and 6. However, cholesterol in the inner leaflet has a downward orientation, with the polar hydroxyl group pointing intracellularly (*η*). A striking observation is that this cholesterol binding site pose was never used as a starting point for simulations and was discovered independent of the pose described in Zhang et al. (2022) (Figure 4—Figure Supplement 1).”

(3) Alternative Possibility: Direct Membrane Access to CRDThe possibility that the CRD extracts cholesterol directly from the membrane outer leaflet is not considered. While the crystal structures place the CRD in a stable pose above the membrane, multiple cryo-EM studies suggest that the CRD is dynamic and adopts a variety of conformations, raising the possibility that the stability of the CRD in the crystal structures is a result of crystal packing and that the CRD may be far more dynamic under more physiological conditions.Recommendation: The authors should explicitly acknowledge and evaluate this potential mechanism and, if feasible, assess its plausibility through MD simulations.

We thank the reviewer for the suggestion. We have addressed this comment by calculating the distance from the lipid headgroups for each lipid in the membrane to the cholesterol binding site. We show that in our study, we do not observe any bending of the CRD over the membrane, precluding any cholesterol from being extracted from the membrane directly.

Added lines: (Paragraph 3, Conclusions):

“An alternative possibility states that the flexibility associated with the CRD would allow it to directly access the membrane, and consequently, cholesterol. In the extensive simulations reported in this study, the binding site of cholesterol in the CRD remains at least 20 Å away from the nearest lipid head group in the membrane, suggesting that such direct extraction and the bending of the CRD do not occur within the timescales sampled (Appendix 2 – Figure 6).

The mechanistic details of this process are still unexplored and form the basis of future work.”

(4) Inconsistent Framing of Study Scope and LimitationsThe discussion contains some contradictory and misleading language. For example, the authors state that "In this study we only focused on the cholesterol movement from the membrane to the CRD binding site," and then several sentences later state that "We outline the entire translocation mechanism from a kinetic and thermodynamic perspective." These statements are at odds. The former appropriately (albeit briefly) notes the limited scope of the modeling, while the latter overstates the generality of the findings.In addition, the authors’ narrow focus on the CRD site constitutes a major caveat to the entire work. It should be acknowledged much earlier in the manuscript, preferably in the introduction, rather than mentioned as an aside in the penultimate paragraph of the conclusion.Recommendation: The authors should clarify the scope of the study and expand the discussion of its limitations. They should explicitly acknowledge that the study models one of several cholesterol access routes and that the findings do not rule out alternative pathways.

We thank the reviewer for the suggestion. We have addressed this comment by explicitly mentioning the scope of the study.

Changes made: (Paragraph 3, Conclusions)

“We outline the entire translocation mechanism from a kinetic and thermodynamic perspective for one of the leading hypotheses for the activation mechanism of SMO.”

(5) Summary:This study has the potential to make a useful contribution to our understanding of cholesterol translocation and SMO activation. However, in its current form, the manuscript presents an overly narrow and, at times, misleading view of the literature and biological models; as such, it is not nearly as impactful as it could be. I strongly encourage the authors to revise the manuscript to include:(1) A more balanced discussion of the CRD vs. TMD binding sites.(2) Acknowledgment of alternative cholesterol access pathways.(3) More comprehensive citation of prior structural and functional studies.(4) Clarification of assumptions and scope.Of note, the above suggestions require little to no additional MD simulations or experimental studies, but would significantly enhance the rigor and impact of the work.

We thank the reviewer for the suggestions. We have taken into account the literature and diverse viewpoints. We have changed the initial discussion and reflected a more general outlook. In the revised version of the manuscript, we have refrained from referring to the CRD site as the orthosteric site. Instead, we refer to it as the CRD sterol-binding site. To better represent the dual-site model, we add further discussion in the Introduction. Through our manuscript, we have avoided using a competitive approach (that one pathway dominates over the other). Instead, we have evaluated both pathways independently and presented a comparative rather than competitive overview of both pathways from our observations. We explicitly mention the scope of the study.